# Nodal-antinodal dichotomy from anisotropic quantum critical continua in holographic models

Ronnie Rodgers[1], Jewel Kumar Ghosh[2,3] and Alexander Krikun[1]

**1** Nordita, KTH Royal Institute of Technology and Stockholm University
Hannes Alfvéns väg 12, SE-106 91 Stockholm, Sweden
**2** Independent University Bangladesh (IUB), Bashundhara RA, Dhaka 1229, Bangladesh
**3** Center for Computational and Data Sciences, Independent University, Bangladesh,
Bashundhara RA, Dhaka 1229, Bangladesh

## Abstract

We demonstrate that the absence of stable quasiparticle excitations on parts of the Fermi surface, similar to the "nodal-antinodal dichotomy" in underdoped cuprate superconductors, can be reproduced in models of strongly correlated electrons defined via a holographic dual. We show analytically that the anisotropy of the quantum critical continuum, which is a feature of these models, may lead to washing out the quasiparticle peak in one direction while leaving it intact in the perpendicular one. The effect relies on the qualitatively different scaling of the self-energy in different directions. Using the explicit example of the anisotropic Q-lattice model, we demonstrate how this effect emerges due to specific features of the near horizon geometry of the black hole in the dual description.

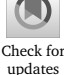
# 1 Introduction

Systems of strongly correlated electrons are not always well-described in terms of stable quasiparticle excitations. An experimental example is the phenomenon of the nodal-antinodal dichotomy in underdoped cuprate superconductors [1], where sharp quasiparticle peaks are observed only on some parts of the Fermi surface. Non-quasiparticle regimes have been demonstrated to exist in theoretical models of strongly correlated electrons constructed using holography, also known as the AdS/CFT correspondence [2–4]. In these models the features of the near horizon geometry of the black hole in the dual description have been shown to govern the self-energy part of the fermionic two-point function in such a way that the inverse quasiparticle lifetime would grow faster than its energy, rendering the stable quasiparticle concept practically useless.

One may describe this holographic effect as the coupling of the would-be quasiparticles to a "quantum critical continuum" bath which mediates their decay, even at small frequencies where the usual elastic decay channels are kinematically forbidden [5]. In systems without rotational symmetry, e.g. due to the underlying crystal lattice, this quantum critical continuum may be present only in some directions in momentum space, and hence it is possible for the quasiparticle excitations to be stable only on some parts of the Fermi surface.

We sketch such a scenario in figure 1. In figure 1a we show the two contributions that we may heuristically think of as combining to give the full fermion spectral function. The plot shows these contributions as functions of momentum in two directions in momentum space, which we label as $k_n$ and $k_a$, where the subscripts stand for (anti)nodal. The blue curve shows the anisotropic quantum critical continuum, which is only present for momentum along one direction ($k_a$). The dashed orange curves show sharp resonance peaks at the Fermi momentum. In figure 1b we show the resulting spectral function. Where the peak overlaps with the quantum critical continuum, as happens along the $k_a$ cut in the figure, interactions between the two broaden the peak, destroying the quasiparticle interpretation. On the other hand, where a peak does not overlap with the continuum it remains sharp and the quasiparticle interpretation is preserved. We refer to these two situations as antinodal and nodal respectively.

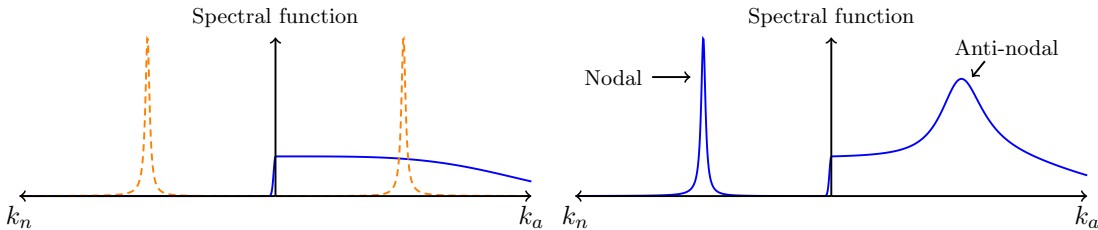

(a) Cuts of contributions to spectral function     (b) Cuts of the full spectral function

Figure 1: Cartoons of the fermion spectral function in the anisotropic strongly correlated systems considered in this work, as a function of momentum in two different directions in momentum space that we label as $k_n$ and $k_a$. **(a):** The spectral function receives contributions from two sources. A quantum critical continuum (solid blue) which exists only for momentum in one direction, which we take to be $k_a$, and resonance peaks (dashed orange). **(b):** When these two contributions are combined any peaks that overlap the continuum are broadened, and thus lose their interpretation as stable quasiparticle excitations.

In this work we provide a comprehensive illustration of how to engineer the above mechanism in holography, showing that it may be realised by an appropriate choice of anisotropic infrared (IR) scaling symmetry. For simplicity we will only focus on cases that preserve a two-

fold rotational symmetry, such that the nodes and antinodes occur along orthogonal directions in momentum space. We comment about a possible generalisation to the experimentally relevant case of four-fold rotational symmetry in section 4.

We start by considering analytically the general case of a holographic model with an anisotropic IR scaling geometry at zero temperature in section 2. Then we specialise to the particular model of the holographic Q-lattice. This model describes a (2+1)-dimensional quantum system with a scalar operator that has a linear profile in one direction $x$. This breaks the translational symmetry in the $x$ direction as well as rotational symmetry. What is important for us is that this model has regimes in which the deep infrared geometry has the aforementioned anisotropic IR scaling behaviour. The self-energy of the quasiparticle excitation then demonstrates nodal behaviour in the $y$ direction and antinodal in the $x$ direction.

In section 3 we consider such Q-lattices at zero temperature and demonstrate the finite width of the quasiparticle peak already at the Fermi surface in the $x$-direction, while in the perpendicular $y$-direction the peak remains sharp and the "quantum critical" self-energy is exponentially suppressed. We also evaluate the fermionic spectral function at the Fermi surface at finite temperature and demonstrate the absence of quasiparticle peaks in certain directions and the behaviour of the corresponding width as a function of temperature. The three appendices are devoted to the details of our analytical treatment (appendix A) and two different numerical methods: shooting at zero temperature, appendix B and pseudospectral relaxation at finite temperature, appendix C. Finally, in appendix D we show some additional plots of numerical results.

## 2 Fermionic Green's functions and anisotropic quantum critical continua

In order to model the effects of the quantum critical continuum we will rely on holographic duality, which postulates the correspondence between strongly correlated quantum systems and classical gravitational theories in an auxiliary curved spacetime with an extra dimension ("the bulk") [6–8].[1] The quantum system under consideration can be taken as defined by its holographic dual, which is the viewpoint we will take throughout this work. Holographic models of condensed matter systems are known to possess an extra "quantum critical" sector in their spectrum, which in particular contributes to the correlation functions and the decay channels of perturbative excitations [11, 12]. In particular, fermionic two-point functions evaluated using holography exhibit non-Fermi liquid self-energies, leading in some cases to the total absence of stable fermionic quasiparticles. As has been shown in refs. [2–4], the behaviour of the imaginary part of the fermionic Green's function at small frequency is controlled by the deep IR, or near-horizon, geometry of the bulk spacetime, which represents this extra "quantum critical contribution". It is therefore sufficient to study the IR asymptotics of the dual geometry in order to conclude whether stable quasiparticles are present in the spectrum or not.

In this work we are interested in cases in which the quantum critical continuum exhibits an anisotropy, which in turn gets imprinted on the fermionic two-point correlation function. Concretely, we will work with bulk gravitational theories containing a $U(1)$ gauge field $A$, and consider metrics and gauge field configurations of the form

$$\mathrm{d}s^2 = -U(r)\,\mathrm{d}t^2 + \frac{\mathrm{d}r^2}{U(r)} + V(r)\,\mathrm{d}x^2 + W(r)\,\mathrm{d}y^2 \,, \qquad A = A_t(r)\,\mathrm{d}t \,, \qquad (1)$$

where $\{t, x, y\}$ are the coordinates in the (2+1)-dimensional strongly correlated quantum

---

[1]For reviews oriented towards condensed matter physics, see refs. [9, 10].

theory, while $r$ parameterises the auxiliary holographic direction. The IR, near-horizon region is $r \to 0$. The strongly correlated system may be thought of as being located at the boundary $r \to \infty$. The gauge field $A$ is associated to the chemical potential in the (2+1)-dimensional system and its associated $U(1)$ current.

The functions $\{U, V, W, A_t\}$ appearing in equation (1) are determined by solving the equations of motion of the gravitational theory with appropriate boundary conditions. Different gravitational theories, and consequently different forms of these functions, are holographically dual to different strongly correlated systems. We will always consider cases in which the geometry (1) is asymptotically four-dimensional anti-de Sitter space (AdS) at large $r$, i.e. $U(r) \approx V(r) \approx W(r) \approx r^2$ as $r \to \infty$, where we employ units in which the asymptotic AdS radius is unity. Apart from this condition, in this section we will keep the functions unfixed. They will then be fixed in section 3 when we consider a concrete model.

In order to holographically obtain the two-point function of a fermionic operator in the (2+1)-dimensional system, one has to solve the Dirac equation in the curved background (1) [2,3],

$$\Gamma^{\underline{a}} e^{\mu}_{\underline{a}} \left( \partial_\mu + \frac{1}{4} \omega_{\mu \underline{bc}} \Gamma^{\underline{bc}} - iqA_\mu \right) \Xi - m\Xi = 0, \tag{2}$$

where $\Xi$ is a four-dimensional Dirac fermion, $\Gamma^{\underline{a}}$ are the flat-space Dirac matrices, $\Gamma^{\underline{ab}} = \frac{1}{2}[\Gamma^{\underline{a}}, \Gamma^{\underline{b}}]$, $e^{\mu}_{\underline{a}}$ is the tetrad associated to the background (1), and $\omega_{\mu \underline{ab}}$ is the spin connection. We use underlines to denote tangent space indices. In equation (2) we have chosen the Dirac fermion to be minimally coupled to the metric and gauge field. We will discuss the role of other interactions in section 4.

The Dirac equation (2) contains two parameters, the mass of the bulk fermion $m$ and its charge $q$. These parametrise the features of the particular fermionic operator dual to $\Xi$, for instance the operator's scaling dimension is $\Delta = m + \frac{3}{2}$, and will affect the size of the Fermi surface. But the phenomenon of the anisotropic destruction of the quasiparticle excitation which we demonstrate here will not depend crucially on the values of $m$ or $q$ and is therefore probe-independent, unlike the other mechanisms proposed earlier in [13, 14].

We now provide a brief description of how fermion Green's functions are computed in holography [15], see appendix A for more details. We wish to compute the Green's function in momentum space, so the first step is to Fourier transform with respect to the field theory directions and consider only a single Fourier mode with frequency $\omega$ and momentum $\vec{k}$,

$$\Xi(t, x, y, r) = e^{-i\omega t + ik_x x + ik_y y} \Xi(r). \tag{3}$$

Let us begin with the case when the fermion has momentum only in the $x$ direction, i.e. $k_y = 0$. In a suitable representation of the Clifford algebra, the Dirac equation (2) may then be partially decoupled into a pair of two-component equations

$$\left[ \sqrt{U(r)} \partial_r + m\sigma^3 - \frac{\omega + qA_t(r)}{\sqrt{U(r)}}(i\sigma^2) \pm \frac{k_x}{\sqrt{V(r)}} \sigma^1 \right] \begin{pmatrix} \chi_\pm(r) \\ i\psi_\pm(r) \end{pmatrix} = 0, \tag{4}$$

where $\sigma^i$ are the Pauli matrices, and $\chi_\pm$ and $\psi_\pm$ are proportional to the components of the four-component spinor $\Xi(r)$ in this representation. Imposing boundary conditions that fix the leading order behaviour of $\psi_\pm$ as $r \to \infty$, the two eigenvalues of the Green's function of the fermionic operator dual to $\Xi$ are given by

$$G_\pm(\omega, k_x, k_y = 0) = -\lim_{r \to \infty} r^{2m} i\chi_\pm/\psi_\pm. \tag{5}$$

For equation (5) to give the retarded (as opposed to advanced, time-ordered etc.) Green's function we must also impose infalling boundary conditions on $(\chi_\pm, \psi_\pm)$ as $r \to 0$ [16, 17].

The spectral function $\rho$ is defined as the imaginary part of the trace of the fermion Green's function, and is thus related to the eigenvalues computed from equation (5) by

$$\rho(\omega, \vec{k}) \equiv \mathrm{Tr}\,\mathrm{Im}\,G(\omega, \vec{k}) = \mathrm{Im}\,G_+(\omega, \vec{k}) + \mathrm{Im}\,G_-(\omega, \vec{k}). \qquad (6)$$

Note that the equation of motion (4) for $k_y = 0$ is independent of the $yy$ metric component $W(r)$. In a similar way, by choosing a different representation one can write down the Dirac equation for $k_x = 0$ which does not depend on $V(r)$. We can therefore obtain qualitatively different fermion Green's functions for momenta in the two orthogonal directions $x$ and $y$ provided $V(r)$ and $W(r)$ are significantly different. We will now analyse the necessary conditions on these functions to obtain the nodal-antinodal behaviour sketched in figure 1.

We will be particularly interested in the behaviour of the fermion spectral function (the imaginary part of the Green's function) near the Fermi surface, i.e. at small frequency and finite momentum. A useful expression for the spectral function (5) in this limit may be developed by artificially dividing the bulk spacetime into an ultraviolet (UV) region $r \geq r_0$ and an IR region $r \leq r_0$, where $r_0$ is a small value of the holographic coordinate [2–4]

$$\mathrm{Im}\,G_\pm(\omega, \vec{k}) = \frac{b_\pm c_\pm - a_\pm d_\pm}{|c_\pm + d_\pm \mathcal{G}_\pm(\omega, \vec{k})|^2}\,\mathrm{Im}\,\mathcal{G}_\pm(\omega, \vec{k}), \qquad \mathcal{G}_\pm(\omega, \vec{k}) \equiv -i\frac{\chi_\pm(r_0)}{\psi_\pm(r_0)}, \qquad (7)$$

where $\{a_\pm, b_\pm, c_\pm, d_\pm\}$ are real functions of $\omega$ and $\vec{k}$ that we will refer to as the matching coefficients. Crucially, these functions are non-zero in the limit $\omega \to 0$. The function $\mathcal{G}_\pm$ is often known as the IR Green's function. Provided $r_0$ is chosen appropriately, namely that it is sufficiently small, both the matching coefficients and the IR Green's function are approximately independent of $r_0$.

Equation (7) shows that the low frequency behaviour of $\mathrm{Im}\,\mathcal{G}_\pm$ determines the low-frequency scaling of the full spectral function. For instance, if the imaginary part of the IR Green's function vanishes at zero frequency as $\mathrm{Im}\,\mathcal{G}_\pm \propto \omega^\nu$ with some positive power $\nu$, then from equation (7) we also have $\mathrm{Im}\,G_\pm \propto \omega^\nu$. Conversely, if $\mathrm{Im}\,\mathcal{G}_\pm$ remains finite as $\omega \to 0$, then so too does $\mathrm{Im}\,G_\pm$. The derivation of equation (7) is reviewed in appendix A.

The low-frequency scaling of the spectral function is thus determined through equation (7) by the behaviours of $(\psi_\pm(r), \chi_\pm(r))$ at small $r$, which are in turn determined by the IR behaviour of the metric and gauge field functions $\{U(r), V(r), W(r), A_t(r)\}$ through equation (4). We will consider cases in which these latter functions behave as power laws at small $r$, as occurs for example in ref. [18],

$$U(r) \approx U_0 r^{\alpha_U}, \quad V(r) \approx V_0 r^{\alpha_V}, \quad W(r) \approx W_0 r^{\alpha_W}, \quad A_t(r) \approx A_0 r^{\alpha_A}, \qquad r \to 0, \qquad (8)$$

where $\alpha_U > 1$ so that the dual strongly correlated system is at zero temperature. In the IR the metric then has a hyperscaling-violating scaling property that can be most clearly seen by defining a new radial coordinate $\zeta = r^{1-\alpha_U}$, in terms of which the IR metric then takes a form similar to that used in refs. [9, 19]. Concretely substituting equation (8) into equation (1) and performing this coordinate transformation, the IR metric becomes (neglecting constant prefactors)

$$ds^2 \approx \zeta^{\theta/\bar{z}}\left(\frac{-dt^2 + d\zeta^2}{\zeta^2} + \frac{dx^2}{\zeta^{2/z_x}} + \frac{dy^2}{\zeta^{2/z_y}}\right), \qquad \zeta = r^{1-\alpha_U}, \qquad (9)$$

where

$$z_x = \frac{2(\alpha_U - 1)}{\alpha_U + \alpha_V - 2}, \quad z_y = \frac{2(\alpha_U - 1)}{\alpha_U + \alpha_W - 2}, \quad \theta = \frac{4(\alpha_U - 2)}{2\alpha_U + \alpha_V + \alpha_W - 4}, \quad \bar{z} \equiv \frac{2 z_x z_y}{z_x + z_y}. \qquad (10)$$

The factor in brackets in equation (9) is manifestly invariant under the rescaling $\{t, \zeta, x, y\} \to \{\lambda t, \lambda \zeta, \lambda^{1/z_x} x, \lambda^{1/z_y} y\}$ with constant $\lambda$, and hence the metric scales as

$\mathrm{d}s^2 \to \lambda^{\theta/\bar{z}} \mathrm{d}s^2$. From the scaling behaviour of $\{t, x, y\}$ we identify $z_{x,y}$ as anisotropic generalisations of the dynamical critical exponent (see also the relevant discussion in [20]).

We identify $\theta$ with an anisotropic version of the hyperscaling violation parameter from the observation that at low non-zero temperatures $T$, the entropy scales as[2]

$$S \propto T^{(2-\theta)/\bar{z}}. \tag{11}$$

In a theory with hyperscaling we would expect from dimensional analysis that $S \propto T^{2/\bar{z}}$. From equation (11) we see that $\theta$ parameterises the deviation from this behaviour. For the isotropic case $z_x = z_y = z$ our definition of $\theta$ reduces to the usual definition used for the hyperscaling violation exponent, for example in refs. [21–24].

Due to the way in which $z_{x,y}$ and $\theta$ appear in equation (9), and due to the way $\alpha_A$ will appear in subsequent equations, it will be convenient to introduce the notation

$$\nu_i \equiv \frac{1}{z_i}, \qquad \nu_\theta \equiv \frac{\theta}{2\bar{z}}, \qquad \nu_A \equiv \frac{2\alpha_A - \alpha_U}{2(\alpha_U - 1)}. \tag{12}$$

From general physical considerations, namely requiring that the entropy in equation (11) does not diverge in the limit $T \to 0$, demanding that the metric in equation (9) satisfies the geometric form of the null energy condition,[3] and requiring the norm of $A_\mu$ not to diverge as $r \to 0$, we find the following constraints on the allowed values of these exponents (see also ref. [20] for similar bounds)

$$-1 + 2\nu_\theta \le \nu_{x,y} \le 1, \qquad \nu_\theta \le \frac{\nu_x + \nu_y}{2} \le 1 - \nu_\theta, \qquad \nu_A \ge 0. \tag{13}$$

Notably, these constraints do not require any of $\nu_{x,y}$ or $\nu_\theta$ to be positive. Indeed, in the next section we will discuss a concrete, top-down holographic model in which $\nu_x < 0$ and $\nu_\theta < 0$, while $\nu_y > 0$.

We now come to the crux of our mechanism to realise the nodal-antinodal behaviour sketched in figure 1. In this IR region, the partially decoupled equations of motion (4) for $k_y = 0$ become

$$\left[\partial_\zeta + \frac{\bar{m}}{\zeta^{1-\nu_\theta}}\sigma^3 - \left(\bar{\omega} + \frac{\bar{q}}{\zeta^{1-2\nu_\theta + \nu_A}}\right)(i\sigma^2) \pm \frac{\bar{k}_x}{\zeta^{1-\nu_x}}\sigma^1\right]\begin{pmatrix}\chi_\pm(\zeta) \\ i\psi_\pm(\zeta)\end{pmatrix} = 0, \tag{14}$$

where the barred parameters are dimensionless versions of $\{m, \omega, q, k_x\}$, defined to eliminate some constant coefficients in the equation of motion.[4] The analogous equations of motion for $k_x = 0$ may be obtained from equation (14) by the substitutions $\bar{k}_x \to \bar{k}_y$ and $\nu_x \to \nu_y$.

Recall that from equation (7) the low-frequency scaling of the fermion spectral function is determined by the solution for $(\chi_\pm, \psi_\pm)$ in the $r \to 0$ region, which in terms of the coordinates used in equation (14) is located at $\zeta \to \infty$. In this limit, the dominant zero-derivative terms in equation (14) are those proportional to $\bar{\omega}$. The behaviour of the IR Green's function depends on how rapidly the remaining terms decay in the same limit:

---

[2]Equation (11) may be proved via the following argument, adapted from ref. [21]. At finite temperature, the thermal entropy $S$ is proportional to the surface area of a horizon located at some $\zeta = \zeta_h$, such that at small temperatures $S \propto \zeta_h^{\theta\bar{z}^{-1} - z_x^{-1} - z_y^{-1}} = \zeta_h^{(\theta-2)/\bar{z}}$. Under the scale transformation, the entropy thus transforms as $S \to \lambda^{-(2-\theta)/\bar{z}}S$. Under the same transformation, temperature transforms as $T \to \lambda^{-1}T$, since $T^{-1}$ should scale in the same way as time. Comparing these scaling behaviours, we arrive at equation (11).

[3]The geometric form of the null energy condition is $R_{\mu\nu}\nu^\mu\nu^\nu \ge 0$, where $R_{\mu\nu}$ is the Ricci tensor and $\nu^\mu$ is a null vector field. It is related to causality in the dual strongly correlated system [25].

[4]Concretely, $\omega = (\alpha_U - 1)U_0\bar{\omega}$, $k_x = (\alpha_U - 1)\sqrt{U_0 V_0}\,\bar{k}_x$, $k_y = (\alpha_U - 1)\sqrt{U_0 W_0}\,\bar{k}_y$, $m = (\alpha_U - 1)\sqrt{U_0}\,\bar{m}$, and $q = (\alpha_U - 1)U_0 A_0^{-1}\bar{q}$.

- If at least one of $v_x > 0$ or $v_\theta > 0$ then equation (14) will contain terms which vanish slower than $\zeta^{-1}$. A perturbative argument shows that the corresponding fermion spectral function will decay exponentially quickly at small frequencies and generic momenta, as $\rho(\omega, k_x, 0) \sim e^{-\alpha/\omega^\beta}$ for some $\alpha, \beta > 0$ [5, 26]. There is no quantum critical continuum.

- If the slowest decaying terms in equation (14) (other than those proportional to $\bar\omega$) decay exactly as $\zeta^{-1}$, i.e. if $v_x \leq 0$ and $v_\theta \leq 0$ with at least one inequality saturated, then we are in the well-studied local quantum critical regime. The spectral function decays as a power law at low frequencies, $\rho(\omega, k_x, 0) \sim \omega^v$ for some $v > 0$ that in general depends on $k_x$ and $m$ [2–4]. There is a quantum critical continuum that may destroy the quasiparticle interpretation, depending on the value of $v$.

- The novel case is when all of the terms in equation (14) (other than those proportional to $\bar\omega$) decay more rapidly than $\zeta^{-1}$. Notice that this requires both $v_x < 0$ and $v_\theta < 0$ (and is independent of the value of $v_A$). We then find that the spectral function at zero frequency is finite, $\rho(\omega, k_x, 0) \sim \omega^0$. We refer once more to appendix A for the proof of this statement. There is a quantum critical continuum of the form sketched in figure 1, which thus always destroys the quasiparticle interpretation.

The same conclusions hold for $\rho(\omega, 0, k_y)$, replacing $v_x$ with $v_y$ in the above. The result is that if we can arrange our scaling exponents such that $v_\theta < 0$, $v_x < 0$, and $v_y > 0$, then there will be a quantum critical continuum present for fermion momentum along $k_x$, but not along $k_y$. In terms of the original scaling exponents introduced in equation (10), these conditions read

$$z_x < 0, \qquad z_y \geq 1, \qquad \theta\left(\frac{1}{z_x} + \frac{1}{z_y}\right) < 0, \tag{15}$$

where we have written $z_y \geq 1$ rather than $z_y > 0$ due to the condition $v_y = z_y^{-1} \leq 1$ in equation (13).[5]

For any background geometry (1) with IR scaling symmetry with exponents satisfying (15), the spectral function of a fermionic operator dual to a Dirac fermion obeying equation (2) will have slices along the momentum axes that look qualitatively like the sketch in figure 1b, with a sharp quasiparticle peak along the $k_y$ axis and a continuum along the $k_x$ axis. The spectral function with both $k_x$ and $k_y$ non-zero must interpolate between these two behaviours. We will show explicit examples in the next section.

Apart from this hard case, characterized by the exponents (15), the nodal-antinodal dichotomy may also be realized in a softer way, when in both directions the Green's function is locally quantum critical (either $v_{x,y}$ or $v_\theta$ is zero), corresponding to the second bullet point above, but the values of the corresponding power law exponents $v$ in the spectral function are different, one being less than, and another bigger than unity. In this case the stable quasiparticles will, again, be well defined in the direction where $v > 1$ and be washed out when $v < 1$. While we don't discuss this softer case in the following, it may also be relevant for some applications.

---

[5]While slightly exotic, negative dynamical exponents have appeared previously in holographic models (see for example ref. [27]) and typically lead to insulating behaviour in the dual strongly correlated system. In this work we focus on the effect of anisotropic scaling with $z_x < 0$ on Fermi surfaces. We leave a full exploration of other physcial consequences, particularly on transport properties, to future work.

# 3 Explicit example: anisotropic Q-lattice

We now provide an explicit demonstration of the effect described in the previous section, using a known holographic model with an anisotropic IR geometry of the form (8), an anisotropic Q-lattice model [18,28].[6] In addition to dynamical gravity and the Maxwell field $A$, this model includes a special complex scalar field $\Phi = \phi\, e^{i\eta}$. We choose this model because of the well-studied dependence of the exponents $\alpha_{U,V,W}$ on the parameters of the model, allowing us to set IR scaling exponents that satisfy the conditions (15). Concretely, we take the model from ref. [18] with parameters $\gamma = 1$, $\alpha = 1$, and $c = 3$, which has bulk gravitational action

$$S = \frac{1}{16\pi G_{\mathrm{N}}} \int \mathrm{d}^4x\, \sqrt{-g}\left[R - \frac{1}{4}\cosh^{1/3}(3\phi)F^2 + 6\cosh\phi - \frac{3}{2}(\partial\phi)^2 - 6\sinh^2\phi\,(\partial\eta)^2\right], \quad (16)$$

where $g$ is the determinant of the metric, $R$ is the Ricci scalar, $F_{\mu\nu} = \partial_\mu A_\nu - \partial_\nu A_\mu$ is the field strength for the Maxwell field, and $G_{\mathrm{N}}$ is Newton's constant.

The equations of motion following from the action (16) admit solutions for the metric and the gauge field of the form (1), with also $\phi = \phi(r)$ and $\eta = px$. This form of $\eta$ breaks both translational symmetry in the $x$ direction and, importantly for us, rotational symmetry in the $(x, y)$-plane. The equations of motion evaluated on this ansatz are written explicitly in appendix B. Near $r = 0$, the metric functions and gauge field obtained from these equations behave as in equation (8), while the amplitude of the scalar field behaves as $e^\phi \approx e^{\phi_0} r^{\alpha_\phi}$. The exponents are [18]

$$\alpha_U = \frac{7}{4}, \qquad \alpha_V = -\frac{1}{4}, \qquad \alpha_W = \frac{3}{4}, \qquad \alpha_A = 1, \qquad \alpha_\phi = -\frac{1}{4}. \quad (17)$$

Substituting these values into equation (10), we find that the anisotropic scaling exponents are $z_x = -3$ and $z_y = 3$. The harmonic mean $\bar{z}$ and the hyperscaling violation exponent $\theta$ are both divergent, in such a way that $\theta/\bar{z} = -1/3$.

Near the boundary at $r \to \infty$, the scalar field and gauge field fall off as

$$\phi(r)\Big|_{r\to\infty} = \lambda r^{-1} + \phi_2 r^{-2} + \mathcal{O}(r^{-3}), \qquad A_t(r)\Big|_{r\to\infty} = \mu + d\, r^{-1} + \mathcal{O}(r^{-2}), \quad (18)$$

where $\{\lambda, \phi_2, \mu, d\}$ are integration constants. The coefficients of the leading-order terms are sources in the dual strongly correlated system and should be fixed by the boundary conditions; $\lambda$ is the source for explicit translational symmetry breaking by the scalar field and $\mu$ is the chemical potential, source of the $U(1)$ charge density. The subleading coefficients are proportional to the vacuum expectation values of the operators sourced by $\lambda$ and $\mu$. In particular, $d$ is proportional to the charge density. These coefficients are determined by the requirement of regularity in the bulk.

We construct solutions with the scaling behaviour (8) at $r \to 0$ and the UV behaviour (18) with fixed $\lambda$ and $\mu$ through a numerical shooting procedure, described in appendix B. After constructing the background solutions, we compute the fermion Green's functions numerically, again using a shooting procedure. Throughout this section, we choose values of the sources such that the two dimensionless ratios in the strongly correlated system are $\lambda/\mu = 1$ and $p/\mu = 0.1$, and we take the bulk fermion $\Xi$ to be massless, i.e. $m = 0$ in equation (2) and to have charge $q = 1$. Some further results for a massive fermion with $m = 1/4$ and with a smaller source $\lambda = 0.1\mu$ are shown in appendix D.

---

[6]The Q-lattice has also been used to study IR anisotropy in ref. [20], in which two non-equivalent scalar fields in two orthogonal directions have been introduced, leading to the same type of anisotropic deep IR metric (8). See other examples of anisotropic holographic geometries in refs. [29–31].

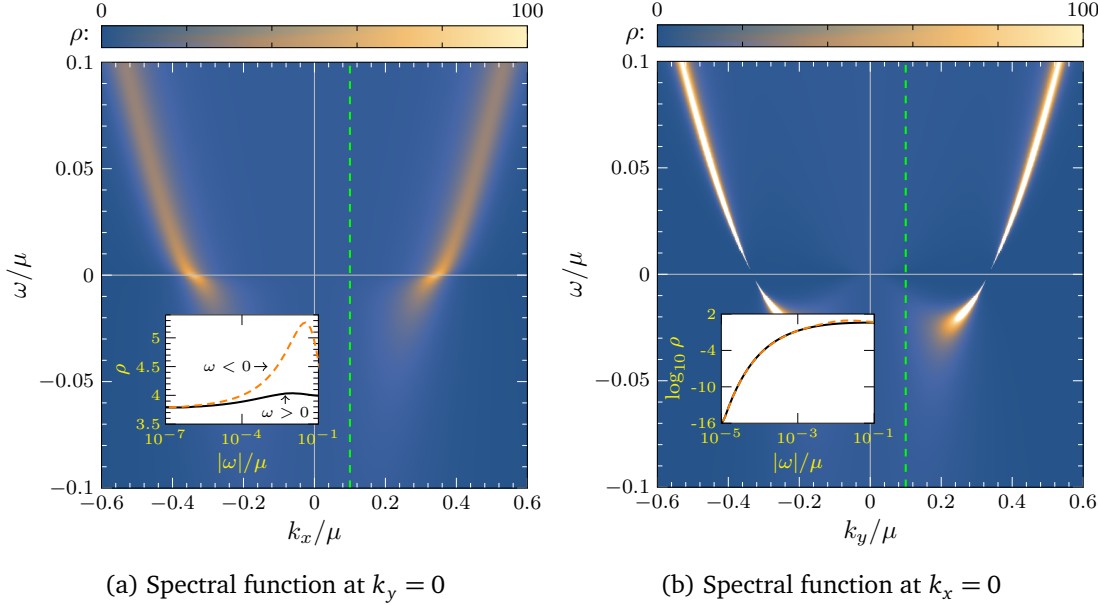

(a) Spectral function at $k_y = 0$

(b) Spectral function at $k_x = 0$

Figure 2: Slices of the fermionic spectral function $\rho(\omega, \vec{k}) = \operatorname{Tr}\operatorname{Im} G(\omega, \vec{k})$ in the anisotropic Q-lattice model (16) at $\lambda = \mu$, $p = 0.1\mu$ and zero temperature, for a bulk fermion with $m = 0$ and $q = 1$. The spectral function is finite at $\omega = 0$ when $\vec{k} = (k_x, 0)$, and vanishes at zero frequency when $\vec{k} = (0, k_y)$. This can clearly be seen in the insets, which show the spectral function along the example constant momentum cuts indicated by the vertical dashed green lines, i.e. at $k_x = 0.1\mu$ in figure 2a and at $k_y = 0.1\mu$ in the inset to figure 2b. The solid black curves in the insets show the spectral function for $\omega > 0$, while the dashed orange curves show the spectral function for $\omega < 0$. The pure white areas in figure 2b have $\rho \geq 100$, outside the scale of the plot.

In figure 2 we show numerical results for slices of the fermion spectral function $\rho(\omega, \vec{k})$ in the $(\omega, k_x)$ and $(\omega, k_y)$ planes. The numerical results agree perfectly with the general prediction made in section 2. Since $\nu_\theta < 0$, $z_x < 0$, and $z_y > 0$, we find that $\rho(\omega, k_x, 0)$ is finite as $\omega \to 0$ for generic $k_x$, while $\rho(\omega, 0, k_y)$ vanishes faster than a power law as $\omega \to 0$. This is most clearly seen in the insets, where we plot the spectral function at fixed momentum $k_x = 0.1\mu$ or $k_y = 0.1\mu$ in Figs. 2a and 2b respectively. As a consequence of this behaviour, the width of the peaks in the spectral function at $k_x \neq 0$ and $k_y = 0$ plotted in figure 2a remain finite at zero frequency, whereas the width of the peaks at $k_x = 0$ and $k_y \neq 0$ plotted in figure 2a shrink to zero.

Given this drastic difference in the behaviour of the spectral function in the two orthogonal directions in momentum space it is particularly interesting to inspect the angular structure of the Fermi surface. This is plotted in figure 3a, where we show the spectral function in the $(k_x, k_y)$ plane, computed at a small imaginary frequency $\omega = 10^{-3} i\mu$ in order to broaden the peaks at $k_x = 0$, making them resolvable numerically. This plot clearly displays an effect similar to nodal-antinodal dichotomy: the Fermi surface is very sharp in the $y$ direction, and barely visible in the $x$ direction.

The precise shape of the spectral function in different directions is shown in figure 3b, where we plot cuts of the Fermi surface at different angles. Concretely, we plot the spectral function at $\omega = 10^{-3} i\mu$ as a function of $|\vec{k}|$, for different values of $\theta = \tan^{-1}(k_y/k_x)$ ranging between $\theta = 0$ and $\theta = \pi/2$. The peak in the $x$ direction ($\theta = 0$) can be discerned, but is an order of magnitude broader than the peak in the $y$ direction ($\theta = \pi/2$).

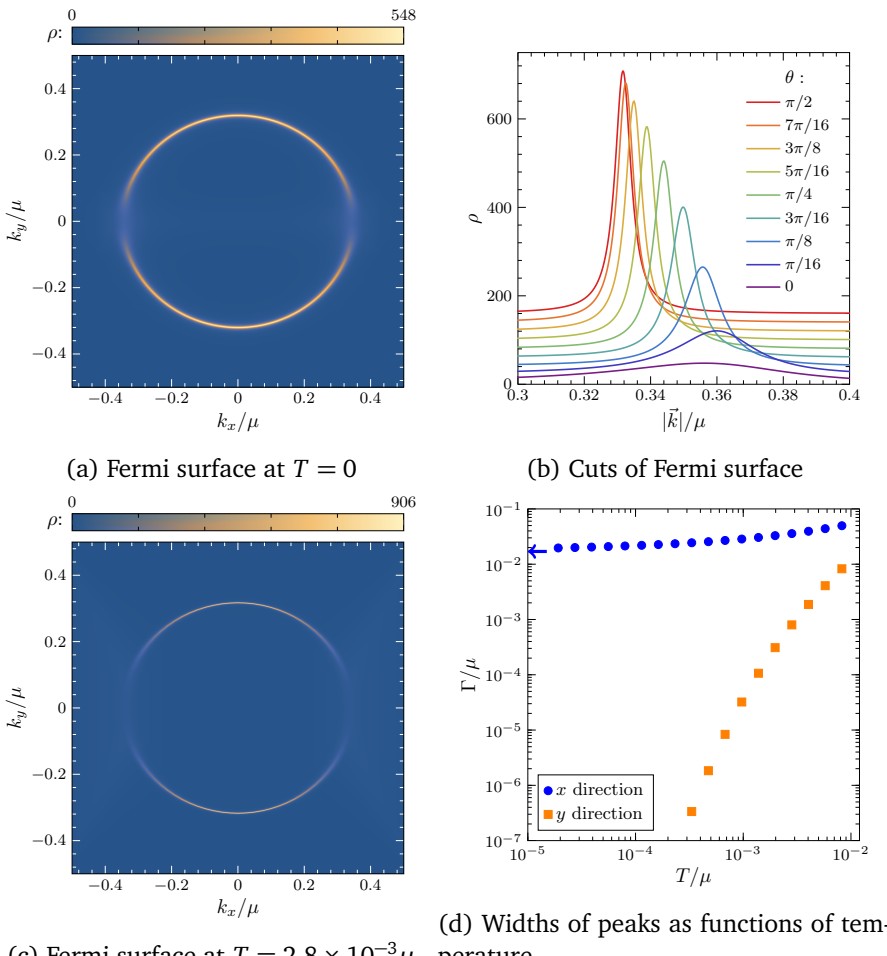

(a) Fermi surface at $T = 0$

(b) Cuts of Fermi surface

(c) Fermi surface at $T = 2.8 \times 10^{-3}\mu$

(d) Widths of peaks as functions of temperature

Figure 3: **(a):** The fermion spectral function as a function of momentum for $\lambda = \mu$, $p = 0.1\mu$, $m = 0$, and $q = 1$ at zero temperature, demonstrating the anisotropic, broken Fermi surface. The spectral function has been evaluated at a small imaginary frequency $\omega = 10^{-3}i\mu$ in order to broaden its peaks, making them resolvable numerically. **(b):** Cuts of the Fermi surface plotted in figure (a), with $k_x = |\vec{k}|\cos\theta$ and $k_y = |\vec{k}|\sin\theta$. Small vertical offsets have been applied to the data to separate the curves for different values of $\theta$. **(c):** The fermion spectral function at $\omega = 0$ and a small non-zero temperature $T = 2.8 \times 10^{-3}\mu$. **(d):** The temperature dependence of the width of the peaks in the spectral function in the $x$ and $y$ directions. The log-log plot clearly demonstrates that the peak in $x$-direction retains a finite width even at zero temperature, while the peak in the $y$-direction sharpens quicker than any power law. Note the similarity to the insets of figure 2. This behaviour demonstrates the qualitatively different features of the Fermi surface in different directions and agrees well with our analytic treatment in section 2. The arrow shows the zero-temperature result $\Gamma \approx 0.017\mu$ in the $x$ direction.

To complete our analysis and make connection to experimental data, we study the widths of the peaks in the spectral function at zero frequency as a function of temperature. We obtain finite temperature gravitational solutions by introducing a black hole horizon at a finite radial coordinate ($r_h \sim 1/T$) in the geometry (1) and solving the Einstein equations numerically as detailed in appendix C. We create a series of anisotropic backgrounds with small temperatures

$T/\mu \in [10^{-5}, 10^{-2}]$ which approach the zero-temperature scaling solution.[7] We then study the evolution of the width of the peaks (full width at half max, $\Gamma$) in the fermionic spectral function computed in these backgrounds, in orthogonal directions on the Fermi surface. One can see from equation (7) that in the vicinity of the peak, i.e. when the denominator has a minimum, the line shape of the momentum distribution curve (MDC) of the spectral density is well approximated by a Lorentzian with full width at half maximum proportional to $\operatorname{Im}\mathcal{G}$. Thus, $\Gamma$ gives us a good indirect measure of the temperature dependence of the quantum critical continuum.

In figure 3d the qualitatively different behaviours are evident: in the $x$ direction the width of the peak approaches a constant when lowering $T$, while in the $y$ direction it decreases faster then any power law (note the logarithmic scales on the plot). We see that as the temperature is lowered, the difference between the two directions grows and the nodal-antinodal dichotomy is enhanced. This behaviour is, again, in good agreement with our analytical deep IR analysis in section 2 and the zero-temperature small frequency scalings obtained above. Finally, we note that the Fermi surface at finite temperature (figure 3c) retains its anisotropic features, which we observed at $T = 0$. Therefore the mechanism which we discuss is not the artifact of the zero-temperature treatment, but can actually affect the phenomenology in the real-life experimental setups.

We stress that in our model this difference in the width of the peaks in different directions can not be solely attributed to the broken translational symmetry and corresponding momentum relaxation, which would indeed broaden the quasiparticle resonance in the $x$ direction, as was shown in refs. [32–37]. As our general analysis in section 2 shows, the Green's functions in the two directions behave qualitatively differently due to the anisotropic features of deep IR geometry. These features are absent when the explicit symmetry breaking is irrelevant in IR as in refs. [32–37]. In particular, for IR-irrelevant deformations the width of the peaks in both directions would become smaller as the temperature is lowered, causing the nodal-antinodal dichotomy effect to disappear at small temperatures. See also the extra data in appendix D, where we demonstrate that our results for the Fermi surface don't depend qualitatively on the value of the explicit symmetry breaking source $\lambda$. Again, as shown by the general analysis in section 2 any bulk theory that has solutions with the appropriate anisotropic IR scaling behaviour will exhibit spectral functions that behave qualitatively similarly to those presented in this section.

Finally, we note that the Fermi surface at finite temperature shown in figure 3c retains the anisotropic features that were observed at $T = 0$. The mechanism that we discuss is thus not an artifact of the zero-temperature treatment, but can actually affect the phenomenology in real-life experimental setups.

## 4 Discussion

In this work we have studied the effects of anisotropy of the quantum critical continuum in systems of strongly correlated and strongly entangled quantum matter defined via their holographic duals. In the dual gravitational description, the features of quantum criticality are encoded in the scaling behaviour of the deep IR geometry. We focus on holographic systems with anisotropic deep IR geometries, with scaling behaviour with exponents obeying the conditions (15), in particular, with a negative dynamical exponent in one direction. We show analytically that in such systems the fermion spectral function behaves at small frequencies qualitatively differently in the two different directions in momentum space, in such a way as to yield an apparent nodal-antinodal dichotomy on the Fermi surface. We check this result by

---

[7]See figure 6, which demonstrates the scaling of entropy in this temperature range.

an explicit numerical calculation in a concrete holographic model.

The crucial part of our finding is that the behaviour we observe is solely dictated by the features of the continuum and the associated scaling exponents (modelled by the IR geometry), and is largely independent of the features of the probe (such as the mass or charge of the probe fermion). In this way it is quite general and would emerge for almost any probe used to describe phenomenological data. We thus might hope that this mechanism for producing a nodal-antinodal dichotomy would work for any type of quantum criticality, whether it can be described by holography or not. Our main statement is therefore that the angular behaviour of the fermionic self-energy, as extracted from e.g. ARPES data in experiments on underdoped cuprates, may reveal important information about the anisotropy of the quantum criticality underlying the system and doesn't have to be explained in terms of fermiology, for example by resonance scattering between the different Fermi surfaces.

Our probe-agnostic treatment goes even further actually. It is easy to see that the deep IR geometry affects bosonic two-point correlation functions in a similar way to fermionic ones, and so one doesn't have to restrict oneself to fermionic probes only. Our mechanism predicts that the anisotropic features of a bosonic probe, which can resolve the angular structure of the finite momentum response, will closely follow the nodal-antinodal dichotomy observed in the fermionic probes, since both are dictated by the same quantum criticality. We suggest that this prediction can be directly checked at available momentum resolved electron energy loss experimental facilities [38, 39] provided the angular dependence is under control. The relations between ARPES and M-EELS measurements have recently been studied in ref. [40] under the assumption of a valid quasiparticle picture. It would be interesting to compare this data to the holographic predictions in detail, so we leave a full exploration of the behaviour of bosonic probes in models of the type considered here to future work.

There are a number of other interesting directions for further research. Although our mechanism for producing a nodal-antinodal dichotomy is probe agnostic in the sense that it is independent of the mass and charge of the probe fermion, the dichotomy may be destroyed by other interactions, depending on how these interactions scale in the IR. To see why, recall that the key to obtaining a quantum critical continuum in the $x$ direction is that, apart from the terms proportional to frequency, the zero-derivative terms in equation (14) decay at least as fast as $\zeta^{-1}$ in the deep IR $\zeta \to \infty$. If we add non-minimal interaction terms to the Dirac equation that decay slower than this, they will destroy the quantum critical continuum.

One interesting and well-motivated non-minimal interaction is a dipole coupling, proportional to $\slashed{F}\Xi$, which can arise in top-down constructions [41, 42] and has been used in holographic models of Mott insulators [43, 44]. It is straightforward to show that including such an interaction adds terms to equation (14) that decay in the deep IR as $\zeta^{-(1+\nu_A)}$. Given the constraint $\nu_A \geq 0$ from equation (13), this interaction decays rapidly enough as $\zeta \to \infty$ that it will not affect the quantum critical continuum.

On the other hand, suppose the gravitational background contains some scalar field $\Phi$. One could add to the Dirac equation (2) a Yukawa-like coupling proportional to $\Phi^n \Xi$ [45], where $n$ is some power that we assume is positive in order to avoid a singularity at $\Phi = 0$. If the scalar field scales as $\Phi \propto r^{\alpha_\Phi}$ in the IR then this adds terms to equation (14) proportional to $\zeta^{-1+\gamma}$ with $\gamma = \nu_\theta + n\alpha_\Phi(2\nu_\theta - 1)$. If $\alpha_\Phi < 0$, i.e. if the scalar field diverges as a power law in the IR, then for sufficiently large $n$ we will find $\gamma > 0$ even when $\nu_\theta < 0$, thus destroying the quantum critical continuum. For example, if we choose $\Phi = e^\phi$ in the Q-lattice model studied in section 3 then we have $\alpha_\Phi = -1/4$ and $\nu_\theta = -1/6$, and consequently $\gamma > 0$ for $n > 1/2$. It would be valuable to perform a complete analysis of possible couplings and their effects on the continuum.

Note also that while in our case the anisotropy of the IR geometry was caused by the explicit Q-lattice, this is not the only way to create such geometries. In particular, holographic models with spontaneous formation of spatially inhomogeneous order would naturally affect the IR geometry in a similar way. This is relevant for various models with spontaneous charge density waves [46] including the holographic model of doped Mott insulators [47] and its homogeneous toy-model version [48].

In another direction, it should be noted that our simple model produces a nodal-antinodal dichotomy with two-fold rotational symmetry, where the "nodes" and "anti-nodes" have an angular separation of $\pi/2$. This is in contrast to known examples of the dichotomy in real materials, where the rotational symmetry is four-fold [1]: the nodes are separated from the anti-nodes by $\pi/4$ (and are also strictly aligned with the crystal lattice). In order to reproduce this phenomenology in a holographic approach one could introduce a periodic 2-dimensional crystal lattice, which will break rotations down to a discrete group and imprint the four-fold anisotropy onto the deep IR geometry. Such periodic lattice constructions have been realised for example in refs. [49–51]. The behaviour of the low frequency, finite momentum two-point functions should then be dictated by the same principles as outlined in section 2. It would be interesting to find such a periodic lattice model with four-fold anisotropic scaling behaviour analogous to the conditions (15),which should then display the desired four-fold nodal-antinodal dichotomy, or its softer version, as described in the end of section 2. The advantage of our simplified model with two-fold rotational symmetry is the possibility for more complete analytic control, drastically improving our understanding of the underlying mechanism for the dichotomy.

# Acknowledgements

We thank Koenraad Schalm, Jan Zaanen, Erik van Heumen, Peter Abbamonte, Blaise Gouteraux and Elias Kiritsis for useful communication and insightful comments. A.K. acknowledges the hospitality of the Lorentz Institute for Theoretical Physics, where the preliminary results of this work have been discussed. The work of A.K is supported by VR Starting Grant 2018-04542 of Swedish Research Council. The numerical computations were enabled by resources provided by the Swedish National Infrastructure for Computing (SNIC), partially funded by the Swedish Research Council through grant agreement no. 2018-05973, at SNIC Science Cloud and PDC Center for High Performance Computing, KTH Royal Institute of Technology. Nordita is supported in part by Nordforsk.

# A    Matching procedure for fermion Green's functions

In this appendix we give details of the matching procedure we use to compute the low-frequency scaling of the fermion Green's function. We will denote the four-dimensional Dirac matrices as $\Gamma^\mu$, they satisfy $\{\Gamma^\mu, \Gamma^\nu\} = 2G^{\mu\nu}$. The tetrad is denoted $e^\mu_{\underline{a}}$, with underlines denoting tangent space indices.[8] The flat space Dirac matrices are then $\Gamma^{\underline{a}} = e^{\underline{a}}_\mu \Gamma^\mu$, satisfying $\{\Gamma^{\underline{a}}, \Gamma^{\underline{b}}\} = 2\eta^{\underline{ab}}$ with $\eta$ the Minkowski metric. We also define projectors $P_\pm = \frac{1}{2}(\mathbb{1} \pm \Gamma^{\underline{r}})$, and

---

[8]Explicitly, we take the non-zero tetrads to be

$$e^r_{\underline{r}} = \sqrt{U(r)}, \qquad e^t_{\underline{t}} = \frac{1}{\sqrt{U(r)}}, \qquad e^x_{\underline{x}} = \frac{1}{\sqrt{V(r)}}, \qquad e^y_{\underline{y}} = \frac{1}{\sqrt{W(r)}}.$$

denote $(\Psi, X) = (P_+ \Xi, P_- \Xi)$. We take the action for the probe fermion to be

$$S = i \int d^4x \sqrt{-G}\, \bar{\Xi}(\slashed{D} - m)\Xi - i \int_{r=r_c} d^3x \sqrt{-\gamma}\, \bar{\Psi}X \,, \tag{A.1}$$

where $r_c$ is a large-$r$ cutoff, $\gamma$ is the induced metric on the hypersurface at $r = r_c$, $\slashed{D} = \Gamma^\mu D_\mu$, with $D_\mu = \partial_\mu + \frac{1}{4}\omega_{\mu ab}\Gamma^{\underline{ab}} - iqA_\mu$ being the covariant derivative.[9] The form of the boundary term in equation (A.1) selects the $r \to \infty$ limits of $\Psi$ and $X$ to be source and vacuum expectation value of the dual fermionic operator, respectively. This is because under a small variation $\Xi \to \Xi + \delta\Xi$, the action (A.1) transforms as

$$\delta S = i \int_{r=r_c} d^3x \sqrt{-\gamma}\left(\bar{X}\,\delta\Psi + \delta\bar{\Psi}X\right), \tag{A.2}$$

when the bulk equations of motion are satisfied. This vanishes if we demand that $\Psi$ is fixed at $r = r_c$.

Evaluated on-shell, the action (A.1) becomes

$$S^\star = -i \int_{r=r_c} d^3x \sqrt{-\gamma}\, \bar{\Psi}X = \int_{r=r_c} d^3x \sqrt{-\gamma}\, \Psi^\dagger \Gamma^{\underline{0}}\mathfrak{g}\Psi \,, \tag{A.3}$$

where we have defined a matrix $\mathfrak{g}(\omega, \vec{k}; r)$ such that $X = -i\mathfrak{g}\Psi$. This matrix may be determined from the equations of motion. Applying the Minkowski space correlator prescription of refs. [16, 17] we then read off the fermion Green's function

$$G(\omega, \vec{k}) = \lim_{r \to \infty} r^{2m}\Gamma^{\underline{0}}\mathfrak{g}(\omega, \vec{k}; r)\,. \tag{A.4}$$

To obtain the factor of $r$ appearing in this limit we have used that the equations of motion discussed below imply that $\Psi \sim r^{m-\frac{3}{2}}$ near the boundary at $r \to \infty$, while $\sqrt{-\gamma} \sim r^3$.

The Dirac equation following from the action (A.1) is $(\slashed{D} - m)\Xi = 0$. Computing the spin connection from the background (1), it is straightforward to show that the covariant derivative takes the form

$$\slashed{D}\Xi = \left[\slashed{\partial} + \Gamma^{\underline{r}}e^r_{\underline{r}}F(r) - iq\slashed{A}\right]\Xi\,, \qquad F(r) = \frac{1}{4}\left(\frac{U'(r)}{U(r)} + \frac{V'(r)}{V(r)} + \frac{W'(r)}{W(r)}\right). \tag{A.5}$$

Defining a new dependent variable through $\Xi = (UVW)^{-1/4}\xi$, we can then eliminate the spin connection from the Dirac equation,

$$\left(\slashed{\partial} - iq\slashed{A} - m\right)\xi = 0\,. \tag{A.6}$$

Applying the projection operators $P_\pm$ to this equation we obtain a pair of coupled equations for $(\psi, \chi) = (P_+\xi, P_-\xi)$. If we also Fourier transform, writing $\psi(r, \tilde{x}) = e^{ip_\mu x^\mu}\psi(r)$ and similar for $\chi$, where $x^\mu = (t, x, y)$ and $p_\mu = (-\omega, k_x, k_y)$, these equations read

$$\left(e^r_{\underline{r}}\partial_r - m\right)\psi + i\left(\slashed{p} - q\slashed{A}\right)\chi = 0\,,$$
$$\left(e^r_{\underline{r}}\partial_r + m\right)\chi - i\left(\slashed{p} - q\slashed{A}\right)\psi = 0\,. \tag{A.7}$$

Let us specialise to the case that the momentum points in the $x$ direction, $p = (\omega, k_x, 0)$. Following [3] we define the projectors $\Pi^x_\pm = \frac{1}{2}\left(\mathbb{1} \pm \Gamma^{\underline{r}}\Gamma^{\underline{t}}\Gamma^{\underline{x}}\right)$. It is then straightforward to show

---

[9]Here we have defined $\Gamma^{\underline{ab}} = \frac{1}{2}\left(\Gamma^{\underline{a}}\Gamma^{\underline{b}} - \Gamma^{\underline{b}}\Gamma^{\underline{a}}\right)$.

that $[\Pi^x_\pm, \not{p}] = [\Pi^x_\pm, \not{A}] = 0$. Defining $\psi_\pm = \Pi^x_\pm \psi$ and $\chi_\pm = \Pi^x_\pm \chi$ and substituting the explict forms of the tetrads, the equations of motion become

$$\sqrt{U}\,\psi'_\pm - m\psi_\pm - i\left(\frac{\omega + qA_t}{\sqrt{U}} \mp \frac{k_x}{\sqrt{V}}\right)\chi_\pm = 0,$$

$$\sqrt{U}\,\chi'_\pm + m\chi_\pm - i\left(\frac{\omega + qA_t}{\sqrt{U}} \pm \frac{k_x}{\sqrt{V}}\right)\psi_\pm = 0. \tag{A.8}$$

Notice that the equations of motion for the $\pm$ sectors are exchanged by sending $k_x \to -k_x$.

The equations of motion (A.7) imply that for $k_x = 0$ we may decompose the matrix $\mathfrak{g}$ as

$$\mathfrak{g} = -\frac{1}{2}(\mathfrak{g}_+ + \mathfrak{g}_-)\Gamma^{\underline{0}} + \frac{1}{2}(\mathfrak{g}_+ - \mathfrak{g}_-)\Gamma^{\underline{1}}, \tag{A.9}$$

where $\mathfrak{g}_\pm$ are the eigenvalues of $\Gamma^{\underline{0}}\mathfrak{g}$. They satisfy $\mathfrak{g}_\pm = -i\chi_\pm/\psi_\pm$. The eigenvalues of the fermion Green's function matrix (A.4) are then

$$G_\pm(\omega, k_x, 0) = \lim_{r\to\infty} r^{2m}\mathfrak{g}_\pm(\omega, \vec{k}; r) = -i \lim_{r\to\infty} r^{2m}\frac{\chi_\pm(r)}{\psi_\pm(r)}. \tag{A.10}$$

The low-frequency scaling of $G_\pm$ may be determined by the behaviour of $\chi_\pm(r)$ and $\psi_\pm(r)$ in the IR region $r \to 0$ [2–4], as we now review. One can solve the first line of equation (A.8) to express $\chi_\pm$ algebraically in terms of $\psi_\pm$,

$$\chi_\pm(r) = -i\frac{\sqrt{U}\psi'_\pm - m\psi_\pm}{U^{-1/2}(\omega + qA_t) \mp V^{-1/2}k_x}. \tag{A.11}$$

This may then be substituted into the second line of equation (A.8) to obtain a second order ODE for $\psi_\pm$ only. The form of this equation will not be important. We just need that all terms in it are real, and that its general solutions take the form

$$\psi_\pm(r) = C_\pm f_\pm(r) - D_\pm g_\pm(r), \tag{A.12}$$

where $C_\pm$ and $D_\pm$ are integration constants, while $f_\pm(r)$ and $g_\pm(r)$ are the so-called normalisable and non-normalisable solutions respectively, with leading-order asymptotics $f_\pm(r) \approx r^m$ and $g_\pm(r) \approx r^{-1-m}$ as $r \to \infty$. Crucially, $f_\pm(r)$ and $g_\pm(r)$ remain real for all $r$, due to the reality of the second order equation for $\psi_\pm$.

Substituting equation (A.12) into equation (A.11) and using these asymptotics, we find that $\chi_\pm(r) \approx iD_\pm r^{-m}$ at large $r$, and thus the Green's function eigenvalues (A.10) are

$$G_\pm(\omega, k_x, 0) = \frac{1 + 2m}{\omega + q\mu - k_x}\frac{D_\pm}{C_\pm}. \tag{A.13}$$

Using equation (A.12) we can express $C_\pm$ and $D_\pm$, and thus the Green's function, in terms of $\psi_\pm$, $f_\pm$, and $g_\pm$ at some arbitrary value of the radial coordinate $r_0$,

$$G_\pm(\omega, k_x, 0) = \frac{\psi_\pm(r_0)f'_\pm(r_0) - \psi'_\pm(r_0)f_\pm(r_0)}{\psi_\pm(r_0)g'_\pm(r_0) - \psi'_\pm(r_0)g_\pm(r_0)} = \frac{a_\pm(r_0) + b_\pm(r_0)\mathcal{G}_\pm(r_0)}{c_\pm(r_0) + d_\pm(r_0)\mathcal{G}_\pm(r_0)}, \tag{A.14}$$

where we have used equation (A.11) to replace $\psi'_\pm$ with $\chi$, defined the IR Green's function $\mathcal{G}(r_0) = -i\chi_\pm(r_0)/\psi_\pm(r_0)$, and defined the matching coefficients

$$a_\pm(r_0) = mf - f'\sqrt{U}, \qquad b_\pm(r_0) = f\left[U^{-1/2}(\omega + q\mu) \mp V^{-1/2}k_x\right],$$

$$c_\pm(r_0) = mg - g'\sqrt{U}, \qquad d_\pm(r_0) = g\left[U^{-1/2}(\omega + q\mu) \mp V^{-1/2}k_x\right], \tag{A.15}$$

where the functions of $r$ on the right-hand sides are to be evaluated at $r_0$. Crucially, all of the matching coefficients are real. The imaginary parts of these eigenvalues are then

$$\operatorname{Im} G_\pm(\omega, k_x, 0) = \frac{b_\pm c_\pm - a_\pm d_\pm}{|c_\pm + d_\pm \mathcal{G}_\pm|^2} \operatorname{Im} \mathcal{G}_\pm, \tag{A.16}$$

where we leave implicit the dependence on $r_0$, $\omega$, and $k_x$ on the right-hand side.

The utility of equation (A.16) is at low frequencies. At leading-order at low frequency, one would be tempted to solve equation (A.8) by setting $\omega = 0$, but this does not quite work due to the factors of $U^{-1/2}$ which diverge as $r \to 0$. Instead, one can set $\omega = 0$ for all $r \geq r_0$ for some small $r_0$. The matching coefficients in equation (A.16) are then approximately independent of frequency, and all $\omega$ dependence in the Green's function comes from the behaviour of the IR Green's function, which can be obtained by solving for $(\psi_\pm(r), \chi_\pm(r))$ only at small $r \leq r_0$.

In this deep IR region, we can approximate the metric functions and gauge field by their power law scaling forms (8). The equations of motion become

$$\psi'_\pm - \frac{m}{\sqrt{U_0}\, r^{\alpha_U/2}} \psi_\pm + i\left(\frac{\omega}{U_0 r^{\alpha_U}} + \frac{q A_0}{U_0\, r^{\alpha_U - \alpha_A}} \mp \frac{k_x}{\sqrt{U_0 V_0}\, r^{(\alpha_U + \alpha_V)/2}}\right)\chi_\pm = 0,$$
$$\chi'_\pm + \frac{m}{\sqrt{U_0}\, r^{\alpha_U/2}} \chi_\pm + i\left(\frac{\omega}{U_0 r^{\alpha_U}} + \frac{q A_0}{U_0\, r^{\alpha_U - \alpha_A}} \pm \frac{k_x}{\sqrt{U_0 V_0}\, r^{(\alpha_U + \alpha_V)/2}}\right)\psi_\pm = 0. \tag{A.17}$$

It is convenient to adopt a new radial coordinate $\zeta = r^{1-\alpha_U}$, and we also define rescaled dimensionless parameters which we will denote with bars, via $\omega = (\alpha_U - 1)U_0 \bar{\omega}$, $k_x = (\alpha_U - 1)\sqrt{U_0 V_0}\,\bar{k}_x$, $k_y = (\alpha_U - 1)\sqrt{U_0 W_0}\,\bar{k}_y$, $m = (\alpha_U - 1)\sqrt{U_0}\,\bar{m}$, and $q = (\alpha_U - 1)U_0 A_0^{-1}\bar{q}$. After these redefinitions, the IR equations of motion (A.17) become

$$\psi'_\pm + \frac{\bar{m}}{\zeta^{1-\nu_\theta}} \psi_\pm + i\left(\bar{\omega} + \frac{\bar{q}}{\zeta^{1-2\nu_\theta + \nu_A}} \mp \frac{\bar{k}_x}{\zeta^{1-\nu_x}}\right)\chi_\pm = 0,$$
$$\chi'_\pm - \frac{\bar{m}}{\zeta^{1-\nu_\theta}} \chi_\pm + i\left(\bar{\omega} + \frac{\bar{q}}{\zeta^{1-2\nu_\theta + \nu_A}} \pm \frac{\bar{k}_x}{\zeta^{1-\nu_x}}\right)\psi_\pm = 0, \tag{A.18}$$

where primes now denote derivatives with respect to $\zeta$. From the constraints on the exponents in equation (13) we see that the powers of $\zeta$ appearing in the denominators of equation (A.18) are all positive.

In the deep IR ($\zeta \to \infty$), the dominant zero-derivative terms in equation (A.18) are those proportional to $\bar{\omega}$. At very large $\zeta$ we can then neglect the terms proportional to $\bar{m}$, $\bar{q}$, and $\bar{k}_x$, and obtain the ingoing solution

$$\psi_\pm \approx e^{i\bar{\omega}\zeta}, \qquad \chi_\pm \approx -e^{i\bar{\omega}\zeta}, \qquad \zeta \to \infty. \tag{A.19}$$

As $\zeta$ is decreased from infinity, other terms in equation (A.18) start to become comparable to the $\bar{\omega}$ terms and must be considered. Which term becomes important first depends on the values of the exponents, and must be treated case by case. Throughout the following we will take $\nu_x < 0$ and $\nu_\theta < 0$. As discussed in section 2, the situations in which at least one of these exponents is non-negative are already well understood.

## A.1 $\bar{k}_x$ term dominant

We first consider the case in which for $\zeta \gg 1$ and $\bar{k}_x \sim \mathcal{O}(1)$:

- $\bar{k}_x \zeta^{\nu_x} \gg \bar{m}\zeta^{\nu_\theta}$, either because $\bar{m} = 0$ or because $\nu_x > \nu_\theta$; and

- $\bar{k}_x \zeta^{\nu_x} \gg \bar{q}\zeta^{2\nu_\theta - \nu_A}$, either because $\bar{q} = 0$ or because $\nu_x > 2\nu_\theta - \nu_A$.

Then at sufficiently large $\zeta$, the IR equations (A.18) are well approximated by

$$\psi'_\pm + i\left(\bar{\omega} \mp \frac{\bar{k}_x}{\zeta^{1-\nu_x}}\right)\chi_\pm = 0, \qquad \chi'_\pm + i\left(\bar{\omega} \pm \frac{\bar{k}_x}{\zeta^{1-\nu_x}}\right)\psi_\pm = 0. \qquad (A.20)$$

It will be convenient to rewrite these equations by defining a rescaled radial coordinate $s = \zeta\left(\bar{\omega}/\bar{k}_x\right)^{1/(1-\nu_x)}$, in terms of which they read

$$\psi'_\pm(s) + i\kappa\left(1 \mp \frac{1}{s^{1-\nu_x}}\right)\chi_\pm(s) = 0, \qquad \chi'_\pm(s) + i\kappa\left(1 \pm \frac{1}{s^{1-\nu_x}}\right)\psi_\pm(s) = 0, \qquad (A.21)$$

where

$$\kappa \equiv \left(\frac{\bar{k}_x}{\bar{\omega}^{\nu_x}}\right)^{1/(1-\nu_x)}. \qquad (A.22)$$

We will be interested in approximate solutions of equation (A.21) when $\bar{k}_x \sim \mathcal{O}(1)$ and $\bar{\omega} \ll 1$. We will be interested in cases where $\nu_x < 0$, and thus from equation (A.22) $\kappa \ll 1$.

We solve equation (A.21) at small $\kappa$ by the following matching procedure. For $s \gg 1$ we can neglect the terms propotional to $s^{-1/(1-\nu_x)}$, and thus the solution obeying ingoing boundary conditions is that given in equation (A.19). In terms of $\kappa$ and $s$ this solution reads

$$\psi_{\pm,R}(s) = e^{i\kappa s}, \qquad \chi_{\pm,R}(s) = -e^{i\kappa s}, \qquad (A.23)$$

where we have added the subscripts $R$ as a reminder that this solution applies to the right of $s = 1$. Near $s \sim \mathcal{O}(1)$ we expect the neglected terms in equation (A.21) proportional to $s^{-1/(1-\nu_x)}$ to become important. However, for $\kappa \ll 1$ they only have a small effect on the solution. To see this, we write the full solution as $\psi_\pm(s) = \psi_{\pm,R}(s) + \delta\psi_\pm(s)$, where $\delta\psi_\pm(s) \to 0$ as $s \to \infty$, and similar for $\chi_\pm$. Substituting into equation (A.21), we then find

$$\delta\psi'_\pm(s) + i\left(1 \mp \frac{1}{s^{1-\nu_x}}\right)\kappa\,\delta\chi_\pm(s) \pm i\kappa\frac{e^{i\kappa s}}{s^{1-\nu_x}} = 0,$$
$$\delta\chi'_\pm(s) + i\left(1 \pm \frac{1}{s^{1-\nu_x}}\right)\kappa\,\delta\psi_\pm(s) \pm i\kappa\frac{e^{i\kappa s}}{s^{1-\nu_x}} = 0. \qquad (A.24)$$

When $\delta\psi_\pm$ and $\delta\chi_\pm$ are small, then the terms proportional to $\kappa\,\delta\psi_\pm$ and $\kappa\,\delta\chi_\pm$ are doubly small in the small-$\kappa$ limit, and may be neglected to leading order in this limit. The approximate solution to equation (A.24) subject to the boundary conditions $\delta\psi_\pm, \delta\chi_\pm \to 0$ as $s \to \infty$ is then

$$\delta\psi_\pm(s) \approx \delta\chi_\pm(s) \approx \pm i\kappa(-i\kappa)^{-\nu_x}\Gamma(\nu_x, -i\kappa s) = \mp\frac{i\kappa s^{\nu_x}}{\nu_x} + \mathcal{O}\left(\kappa^{1+|\nu_x|}\right), \qquad (A.25)$$

where $\Gamma$ is the incomplete Gamma function, and on the right-hand side we have expanded for fixed $s$ and small $\kappa$. The right-hand side vanishes in the limit $\kappa \to 0$, and thus $\psi_{\pm,R}$ and $\chi_{\pm,R}$ indeed provide good approximate around $s = 1$ at low frequencies. This also justifies a posteriori our neglect of the terms in equation (A.24) proportional to $\kappa\,\delta\psi_\pm$ and $\kappa\,\delta\chi_\pm$. On the other hand, this approximation breaks down at $s \ll 1$, where the incomplete gamma functions grow very large.

For $s \ll 1$, the terms in equation (A.21) proportional to $s^{-(1-\nu_x)}$ become dominant, and we obtain the approximate solution

$$\psi_{\pm,L}(s) \approx a_\pm \exp\left(\frac{\kappa}{\nu_x}s^{\nu_x}\right) + b_\pm \exp\left(-\frac{\kappa}{\nu_x}s^{\nu_x}\right),$$
$$\chi_{\pm,L}(s) \approx \mp ia_\pm \exp\left(\frac{\kappa}{\nu_x}s^{\nu_x}\right) \pm ib_\pm \exp\left(-\frac{\kappa}{\nu_x}s^{\nu_x}\right), \qquad (A.26)$$

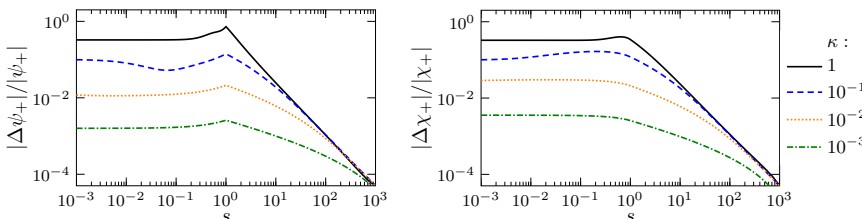

Figure 4: Demonstration of the validity of the matching procedure at small $\kappa$. We plot the error in $\psi_+$ and $\chi_+$ obtained from the matching calculation relative to a numerical solution of equation (A.21) with ingoing boundary conditions $\psi_+ \approx e^{i\kappa s}$ and $\chi_+ \approx e^{-i\kappa s}$ at large $s$. In other words, what is plotted is the absolute value of the difference between the matching and numerical solutions, divided by the numerical solution. For concreteness we take $\nu_x = -1/3$ (the same value used in section 3) and $s_m = 1$. The relative error clearly becomes very small at small $\kappa$.

where we have added the subscripts $L$ as a reminder that this solution applies to the left of $s = 1$. An argument almost identical to the one we made for $(\psi_{\pm,R}, \chi_{\pm,R})$ shows that $(\psi_{\pm,L}, \chi_{\pm,L})$ are good approximations to the full solution of equation (A.21) near $s \sim \mathcal{O}(1)$ when $\kappa \ll 1$.

We fix the integration constants $a_\pm$ and $b_\pm$ in equation (A.26) by matching to equation (A.23) at some $s_m$ $\mathcal{O}(1)$, near which both solutions provide good approximations. Setting $\psi_{\pm,L}(s_m) = \psi_{\pm,R}(s_m)$ and $\chi_{\pm,L}(s_m) = \chi_{\pm,R}(s_m)$, we can solve for the integration constants to find

$$
\begin{aligned}
a_\pm &= \frac{1}{\sqrt{2}} \exp\left( i\kappa s_m - \frac{\kappa}{\nu_x} s_m^{\nu_x} \mp \frac{i\pi}{4} \right) = \frac{e^{\mp i\pi/4}}{\sqrt{2}} + \mathcal{O}(\kappa), \\
b_\pm &= \frac{1}{\sqrt{2}} \exp\left( i\kappa s_m + \frac{\kappa}{\nu_x} s_m^{\nu_x} \pm i\frac{\pi}{4} \right) = \frac{e^{\pm i\pi/4}}{\sqrt{2}} + \mathcal{O}(\kappa).
\end{aligned}
\tag{A.27}
$$

As a check of our matching procedure we plot the relative errors $|\psi_{+,m} - \psi_+|/|\psi_+|$ and $|\chi_{+,m} - \chi_+|/|\chi_+|$ for $\nu_x = -1/3$ and $s_m = 1$, and sample values of $\kappa$ in figure 4, where $(\psi_+, \chi_+)$ are a numerical solution of equation (A.21) with ingoing boundary conditions, while $(\psi_{+,m}, \chi_{+,m})$ are matched approximations, given by equation (A.23) for $s > 1$ and equation (A.26) for $s < 1$, with coefficients (A.27). The relative error indeed becomes very small at small $\kappa$.

Returning to the radial coordinate $\zeta = s(\bar{k}_x/\bar{\omega})^{1/(1-\nu_x)}$, we have found the following approximate solution to equation (A.20), obeying ingoing boundary conditions at $\zeta \to \infty$ and valid at small $\bar{\omega}$ and fixed $\bar{k}_x$,

$$
\begin{aligned}
\psi_\pm(\zeta) &= \begin{cases} e^{i\bar{\omega}\zeta}, & \zeta > \zeta_m, \\ \sqrt{2}\cosh\left( \frac{\bar{k}_x}{\nu_x} \zeta^{\nu_x} - \frac{i\pi}{4} \right), & \zeta < \zeta_m, \end{cases} \\
\chi_\pm(\zeta) &= \begin{cases} -e^{i\bar{\omega}\zeta}, & \zeta > \zeta_m, \\ \mp i\sqrt{2}\sinh\left( \frac{\bar{k}_x}{\nu_x} \zeta^{\nu_x} - \frac{i\pi}{4} \right), & \zeta < \zeta_m, \end{cases}
\end{aligned}
\tag{A.28}
$$

where $\zeta_m = s_m(\bar{k}_x/\bar{\omega})^{1/(1-\nu_x)}$.

As $\zeta$ is decreased from infinity, a point will eventually be reached where equation (A.20) ceases to provide a good approximation to the full equations of motion, and thus the matching solution (A.28) will cease to provide a good approximate solution. This is both because of the

terms we have neglected from equation (A.18) and because the metric functions and gauge field will depart from their scaling forms (8). At low frequencies the approximate location at which this occurs will be independent of frequency, being determined by the relative size of the neglected terms and the terms proportional to $\bar{k}_x$ in equation (A.20).

We can thus choose to evaluate the IR Green's function $\mathcal{G}_\pm = -i\chi_\pm(\zeta_0)/\psi_\pm(\zeta_0)$ at some frequency-independent $\zeta_0$ satisfying $1 \ll \zeta_0 \ll \zeta_m$ (note that $\zeta_m \to \infty$ as $\bar{\omega} \to 0$), where we choose $\zeta_0$ to lie within the range of validity of the matching solution. We can then use equation (A.28) to evaluate the IR Green's function, obtaining

$$\mathcal{G}_\pm(\omega, k_x, 0) \approx i, \qquad \bar{\omega} \ll 1. \tag{A.29}$$

Recalling that the low frequency scaling of $\operatorname{Im}\mathcal{G}_\pm$ determines the low frequency scaling of the imaginary part of the full Green's function, we find that the spectral function tends to a non-zero constant at small frequency.

## A.2 $\bar{q}$ term dominant

Now we consider the case in which the first subleading terms in equation (A.18) at large $\zeta$ are the ones proportional to $\bar{q}$. This will occur if $\nu_x < 2\nu_\theta - \nu_A$ and $\bar{m} = 0$.[10] In this case, at sufficiently large $\zeta$ the IR equations (A.18) will be well approximated by

$$\psi'_\pm + i\left(\bar{\omega} + \frac{\bar{q}}{\zeta^{1-2\nu_\theta+\nu_A}}\right)\chi_\pm = 0, \qquad \chi'_\pm + i\left(\bar{\omega} + \frac{\bar{q}}{\zeta^{1-2\nu_\theta+\nu_A}}\right)\psi_\pm = 0. \tag{A.30}$$

These can be solved exactly. With ingoing boundary conditions we have

$$\psi_\pm(\zeta) = \exp\left(i\bar{\omega}\zeta - i\frac{\bar{q}\zeta^{-(\nu_A-2\nu_\theta)}}{\nu_A - 2\nu_\theta}\right), \qquad \chi_\pm(\zeta) = -\exp\left(i\bar{\omega}\zeta - i\frac{\bar{q}\zeta^{-(\nu_A-2\nu_\theta)}}{\nu_A - 2\nu_\theta}\right), \tag{A.31}$$

valid over the region for which equation (A.30) provides a good approximation to the full equations of motion. The eigenvalues of the IR Green's function are $\mathcal{G}_\pm = -i\chi_\pm(\zeta_0)/\psi_\pm(\zeta_0)$, where $\zeta_0$ is some large value of $\zeta$ within this region. From equation (A.31) we find

$$\mathcal{G}_\pm(\omega, k_x, 0) \approx i, \qquad \bar{\omega} \ll 1. \tag{A.32}$$

Again, the fermion spectral function tends to a non-zero value at zero frequency.

## A.3 $\bar{m}$ term dominant

Finally, we consider the case in which for $\zeta \gg 1$ and $\bar{k}_x \sim \mathcal{O}(1)$ that $\bar{m}\zeta^{\nu_\theta} \gg \bar{k}_x\zeta^{\nu_x}$. This requires $\nu_\theta > \nu_x$. The calculation is very similar to that of section A.1, so we will be brief with the details. At sufficiently large $\zeta$, the IR equations (A.18) are well approximated by

$$\psi'_\pm + \frac{\bar{m}}{\zeta^{1-\nu_\theta}}\psi_\pm + i\bar{\omega}\chi_\pm = 0, \qquad \chi'_\pm - \frac{\bar{m}}{\zeta^{1-\nu_\theta}}\chi_\pm + i\bar{\omega}\psi_\pm = 0. \tag{A.33}$$

It will be convenient to rewrite these equations by defining a rescaled radial coordinate $s = \zeta(\bar{\omega}/\bar{m})^{1/(1-\nu_\theta)}$ (note that this is a different $s$ from section A.1), in terms of which they read

$$\psi'_\pm(s) + \frac{\mathfrak{m}}{s^{1-\nu_\theta}}\psi(s) + i\mathfrak{m}\chi(s) = 0, \qquad \chi'_\pm(s) - \frac{\mathfrak{m}}{s^{1-\nu_\theta}}\chi(s) + i\mathfrak{m}\psi(s) = 0, \tag{A.34}$$

---

[10]Notice that for non-zero mass and $\nu_\theta \leq 0$, the mass term always dominates over the charge term, $\bar{m}\zeta^{\nu_\theta} \gg \bar{q}\zeta^{2\nu_\theta-\nu_A}$ at large $\zeta$, due to the requirement $\nu_A \geq 0$ in equation (13).

where

$$\mathfrak{m} = \left(\frac{m}{\omega^{\nu_\theta}}\right)^{1/(1-\nu_\theta)} \ll 1. \tag{A.35}$$

At large $s$ we neglect the terms in equation (A.34) proportional to $s^{-1/(1-\nu_\theta)}$, obtaining the ingoing solution

$$\psi_{\pm,R}(s) = e^{i\mathfrak{m}s}, \qquad \chi_{\pm,R}(s) = -e^{i\mathfrak{m}s}. \tag{A.36}$$

On the other hand, at small $s$ the terms in equation (A.34) proportional to $s^{-1/(1-\nu_\theta)}$ are dominant, and we obtain the approximate solution

$$\psi_{\pm,L}(s) = a_\pm e^{-\mathfrak{m}s^{\nu_\theta}/\nu_\theta}, \qquad \chi_{\pm,L}(s) = b_\pm e^{\mathfrak{m}s^{\nu_\theta}/\nu_\theta}, \tag{A.37}$$

where $a_\pm$ and $b_\pm$ are integration constants.

An argument similar to the one made in section A.1 shows that both $(\psi_{\pm,L}, \chi_{\pm,L})$ and $(\psi_{\pm,R}, \chi_{\pm,R})$ are good approximate solutions at $s \sim \mathcal{O}(1)$ when $\mathfrak{m} \ll 1$. We can thus fix the integration constants by setting $\psi_{\pm,L}(s_m) = \psi_{\pm,R}(s_m)$ and $\chi_{\pm,L}(s_m) = \chi_{\pm,R}(s_m)$. This yields $a_\pm = 1$ and $b_\pm = -1$ to leading order at small $\mathfrak{m}$ (and thus at small $\bar{\omega}$). In terms of the radial coordinate $\zeta$, the approximate matching solution is then

$$\psi_\pm(\zeta) = \begin{cases} e^{i\bar{\omega}\zeta}, & \zeta > \zeta_m, \\ e^{\bar{m}\zeta^{\nu_\theta}/\nu_\theta}, & \zeta < \zeta_m, \end{cases} \qquad \chi_\pm(\zeta) = \begin{cases} e^{i\bar{\omega}\zeta}, & \zeta > \zeta_m, \\ -e^{\bar{m}\zeta^{\nu_\theta}/\nu_\theta}, & \zeta < \zeta_m, \end{cases} \tag{A.38}$$

where $\zeta_m = s_m(\bar{m}/\bar{\omega})^{1/(1-\nu_\theta)}$. The IR Green's function is $\mathcal{G}_\pm = -i\chi_\pm(\zeta_0)/\psi_\pm(\zeta_0)$ for some $\zeta_0$ satisfying $1 \ll \zeta_0 \ll \zeta_m$. Evaluating this using equation (A.38), we find that at leading order at small frequencies

$$\mathcal{G}_\pm(\omega, k_x, 0) \approx i, \qquad \bar{\omega} \ll 1. \tag{A.39}$$

# B  Numerical shooting method at zero temperature

In this appendix we describe the numerical shooting procedure we used to obtain zero-temperature solutions to the gravitational model (16). It will be convenient to work in Fefferman-Graham gauge for the metric,

$$ds^2 = \frac{dz^2}{z^2} - U(z)\,dt^2 + V(z)\,dx^2 + W(z)\,dy^2, \tag{B.1}$$

where $z$ (not to be confused with a dynamical exponent) is related the radial coordinate $r$ of equation (1) by $dz/z = -dr/\sqrt{U(r)}$. With a slight abuse of notation we use $U(z)$ to denote $U(r(z))$ in equation (B.1), and similar for $V$ and $W$. The conformal boundary is located at $z = 0$. For the other fields we make the ansatz $A = A_t(z)\,dt$, $\phi = \phi(z)$, and $\chi = px$ for some constant $p$. This ansatz automatically solves the Euler-Lagrange equation for $\chi$, while

the remaining equations of motion take the form

$$\frac{V'V''}{V^2} - \frac{W'W''}{W^2} + \frac{1}{2}\left(\frac{W'^2}{W^2} - \frac{V'^2}{V^2}\right) + \frac{3W}{2V}\left(\frac{V}{W}\right)'\phi'^2 + \frac{6p^2\mathcal{S}^2(VW)'}{V^2W} = 0\,, \quad \text{(B.2a)}$$

$$\frac{(VW)'(W'V')'}{V^2W^2} - \frac{V'^2W'^2}{V^2W^2} + \frac{12p^2\mathcal{S}^2W'^2}{VW^2} + \frac{3V'W'}{VW}\phi'^2 + \left(\frac{V'^2}{V^2} + \frac{W'^2}{W^2}\right)\left(\frac{Q^2}{2VW\tilde{\mathcal{C}}} - 6\tilde{\mathcal{C}}\right) = 0\,, \quad \text{(B.2b)}$$

$$\frac{(VW)'}{VW}(\phi'' + 2\mathcal{S}) + \frac{3}{2}\phi'^2 + \frac{1}{2}\left(\frac{V'^2}{V^2} + \frac{V'W'}{VW} + \frac{W'^2}{W^2} + 12\mathcal{C}\right)\phi'$$
$$-\frac{p^2}{V}\left(\frac{4(VW)'\mathcal{C}}{VW} + 6\mathcal{S}\phi'\right)\mathcal{S} - \frac{Q^2}{2\tilde{\mathcal{C}}VW}\phi' + \frac{Q^2(VW)'}{6V^2W^2}\frac{\tilde{\mathcal{S}}}{\tilde{\mathcal{C}}^4} = 0\,, \quad \text{(B.2c)}$$

$$(VW)'\frac{U'}{U} + V'W' + \frac{Q^2}{\tilde{\mathcal{C}}} + 12p^2W\mathcal{S}^2 - 3VW\left(\phi'^2 + 4\mathcal{C}\right) = 0\,, \quad \text{(B.2d)}$$

$$A_t' - \frac{Q}{\tilde{\mathcal{C}}}\sqrt{\frac{U}{VW}} = 0\,, \quad \text{(B.2e)}$$

where we have employed the notation $\mathcal{C} \equiv \cosh\phi$, $\mathcal{S} \equiv \sinh\phi$, $\tilde{\mathcal{C}} \equiv [\cosh(3\phi)]^{1/3}$, and $\tilde{\mathcal{S}} \equiv \sinh(3\phi)$. To obtain the first-order equation (B.2e) we have integrated the Euler-Lagrange equation for $A_t$ once, with $Q$ the corresponding integration constant. Equations (B.2) are invariant under the separate rescalings $(U, A_t^2) \to \Omega_t(U, A_t^2)$, $(V, p^2, Q^2) \to \Omega_x(V, p^2, Q^2)$ and $(W, Q^2) \to \Omega_y(W, Q^2)$, with constant scale factors $\Omega_i$, reflecting the freedom to change the units in which we measure the field theory coordinates $(t, x, y)$.

Near the conformal boundary at $z = 0$, solutions to the equations of motion (B.2) have asymptotic expansions of the form

$$V(z) = \frac{c_V}{z^2}\left[1 - \frac{3}{8}\lambda^2 z^2 - V_3 z^3 + \mathcal{O}(z^4)\right]\,, \quad \text{(B.3a)}$$

$$W(z) = \frac{c_W}{z^2}\left[1 - \frac{3}{8}\lambda^2 z^2 - W_3 z^3 + \mathcal{O}(z^4)\right]\,, \quad \text{(B.3b)}$$

$$\phi(z) = \lambda z + \phi_2 z^2 + \mathcal{O}(z^3)\,, \quad \text{(B.3c)}$$

$$U(z) = \frac{c_U}{z^2}\left[1 - \frac{3}{8}\lambda^2 z^2 - (V_3 + W_3 - 2\lambda\phi_2)z^3 + \mathcal{O}(z^4)\right]\,, \quad \text{(B.3d)}$$

$$A_t(z) = \mu - \frac{Q}{\sqrt{c_V c_W}}z + \mathcal{O}(z^2)\,, \quad \text{(B.3e)}$$

with integration constants $(c_{U,V,W}, V_3, W_3, \lambda, \phi_2, \mu)$. We impose boundary conditions such that $c_{U,V,W} = 1$. The non-normalisable coefficients $\lambda$ and $\mu$ correspond to sources in the dual field theory and must also be fixed.

The reamining four integration constants $(V_3, W_3, \phi_2)$, as well as $Q$, are then determined by the requirement of regularity in the bulk. At some $z = z_0$ there is an extremal horizon, near which regular solutions to equation (B.2) have expansions that are most conveniently written

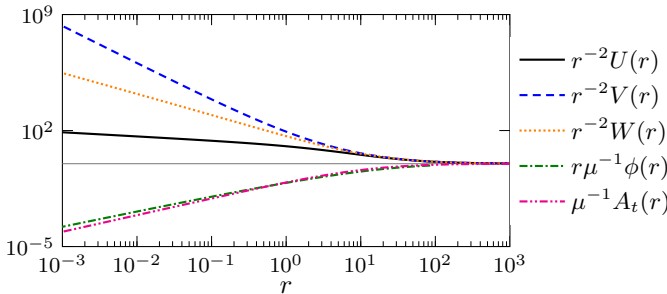

Figure 5: Example numerical background solution obtained from our shooting procedure, for $\lambda = \mu$ and $p = 0.1\mu$. The plotted functions all tend to unity in the UV ($r \to \infty$), indicated by the horizontal grey line.

in powers of $\ell \equiv \log(z_0/z) \ll 1$, of the form

$$V(z) \approx \frac{100 p^2}{3\ell^2} \left[ 1 - 0.00123\, \ell^4 + c_1 \ell^{\gamma_1} + c_2 \ell^{\gamma_2} + \mathcal{O}(\ell^8) \right], \tag{B.4a}$$

$$W(z) \approx \frac{9 Q^2 \ell^6}{256000 \times 2^{2/3} p^2} \left[ 1 - 0.00808\, \ell^4 - 1.29\, c_1 \ell^{\gamma_1} + 0.182\, c_2 \ell^{\gamma_2} + \mathcal{O}(\ell^8) \right], \tag{B.4b}$$

$$e^{\phi(z)} \approx \frac{80}{3\ell^2} \left[ 1 - 0.00199\, \ell^4 - 3.10\, c_1 \ell^{\gamma_1} - 0.0556\, c_2 \ell^{\gamma_2} + \mathcal{O}(\ell^8) \right], \tag{B.4c}$$

$$U(z) \approx U_0 \ell^{14} \left[ 1 + 0.00757\, \ell^4 - 0.152\, c_1 \ell^{\gamma_1} - 0.636\, c_2 \ell^{\gamma_2} + \mathcal{O}(\ell^8) \right], \tag{B.4d}$$

$$A_t(z) \approx 2^{-11/6} \sqrt{\frac{3 U_0}{5}} \ell^8 \left[ 1 + 0.00695\, \ell^4 + 0.482\, c_1 \ell^{\gamma_1} + 0.450\, c_2 \ell^{\gamma_2} + \mathcal{O}(\ell^8) \right], \tag{B.4e}$$

with exponents $\gamma_1 = 4\left(\sqrt{5} - 1\right) \approx 4.94$ and $\gamma_2 = 4(\sqrt{69} - 3)/3 \approx 7.08$, and three integration constants $(U_0, c_1, c_2)$. The decimals in these expansions represent approximations to exact numbers determined by the equations of motion.

In practice we construct numerical solutions to the equations of motion (B.2) by shooting from the near horizon region as follows. First, we fix $Q = p = z_0 = 1$ and choose some seed values of $c_1$ and $c_2$. We then use equation (B.4) to set boundary conditions near $z = z_0$, using which we solve equations (B.2a), (B.2b), and (B.2c) to obtain $(V, W, \phi)$. From the small-$z$ behaviour of the solutions for $V$ and $W$ we read off the UV coefficients $c_{V,W}$. We then obtain new solutions with the correct boundary conditions $c_V = c_W = 1$ using the rescaling symmetries mentioned under equations (B.2). Notice that the rescaling will generically set $Q \neq 1$ and $p \neq 1$.

We now substitute the (rescaled) solutions for $(V, W, \phi)$ into equation (B.2d) and numerically solve for $U$, using equation (B.3d) with $c_U = 1$ to set boundary conditions at small $z$. For the behaviour of this solution near $z = z_0$ we then read off the coefficient $U_0$ appearing in equation (B.4). Finally, we substitute our solutions for $(U, V, W, \phi)$ into equation (B.2e), which we then solve for $A_t$ using the boundary condition that $A_t(z = z_0) = 0$.

The output of this procedure is a solution to the equations of motion (B.2) with a priori unknown values of the dimensionless ratios $\lambda/\mu$ and $p/\mu$. We then use the Newton-Raphson procedure to find the values of the IR coefficients $c_{1,2}$ the yield the desired values of $\lambda/\mu$ and $p/\mu$. For example, in figure 5 we plot the resulting solution with $\lambda = \mu$ and $p = 0.1\mu$, showing the power law scalings in the deep IR ($r \to 0$) and the UV asymptotics (18) and $U(r) \approx V(r) \approx W(r) \approx r^2$ at $r \to \infty$.

Having computed the backgrounds, we obtain the fermion Green's functions numerically as follows, closely following refs. [52–56]. We solve directly for the matrix $\mathfrak{g}$ introduced in

equation (A.3). To do so we derive use the definition[11] $\chi = -i\mathfrak{g}\psi$ to derive a first order ODE for $\mathfrak{g}$ from equation (A.7),

$$e^r_{\underline{r}}\partial_r\mathfrak{g} - 2m\mathfrak{g} + \not{p} - q\not{A} - \mathfrak{g}(\not{p} - q\not{A})\mathfrak{g} = 0.\tag{B.5}$$

Decomposing[12] $\mathfrak{g}(r) = \mathfrak{g}_0(r)\Gamma^{\underline{0}} + \mathfrak{g}_1(r)\Gamma^{\underline{1}} + \mathfrak{g}_2(r)\Gamma^{\underline{2}}$ we obtain three coupled equations for the coefficients $\mathfrak{g}_{0,1,2}(r)$, which we solve numerically, integrating from small $r$ where we impose the boundary conditions $\mathfrak{g}_0(0) = i$ and $\mathfrak{g}_1(0) = \mathfrak{g}_2(0) = 0$. We then obtain the fermion Green's function from the large-$r$ behaviour of the numerical solution using equation (A.4).

## C  Numerical pseudospectral method at finite temperature

At finite temperature we use the pseudospectral relaxation method in order to evaluate the Q-lattice background solutions as well as fermionic spectral functions. This is different from the numerical shooting method, described in appendix B, which we use at zero temperature. The reason for choosing a different numerical method here is two-fold. Firstly, the relaxation method allows for a more convenient treatment of the boundary values both in IR and UV, and since it can be used at finite temperature there is no reason not to enjoy this convenience. Secondly, the matching between the two completely different methods allows us to cross check our results very efficiently.

In order to implement the pseudospectral relaxation (which we do along the lines of [57], see also [58, 59]) one first has to put the Einstein equations in elliptic form. This is achieved by the DeTurk trick discussed in [60–62], which allows one to fix the gravitational gauge dynamically. As a result of this procedure, we obtain four second order elliptic equations for the four components of the diagonal 4D metric in the bulk. In particular we parametrize the Q-lattice background metric by the ansatz:

$$ds^2 = \frac{1}{u^2}\left(-T(u)f(u)dt^2 + \frac{U(u)}{f(u)}du^2 + W_1(u)dx^2 + W_2(u)dy^2\right),\tag{C.1}$$

$$f(u) = (1-u)P(u), \qquad P(u) = 1 + u + u^2 - \frac{\tilde{\mu}^2}{4}u^3.\tag{C.2}$$

Note that we use the freedom in defining the coordinate $u$ in order to set the horizon radius to $u_h = 1$. The parameter $\tilde{\mu}$ sets the temperature of this background as

$$T = \frac{12 - \tilde{\mu}^2}{16\pi}.\tag{C.3}$$

We use the Reissner-Nordström metric $T(u) = U(u) = W_1(u) = W_2(u) = 1$ as a reference metric in the DeTurk procedure.

The gauge field, which vanishes at the horizon, and the scalar are parametrized as

$$A = a(u)(1-u)dt,\tag{C.4}$$

$$\phi = u\psi(u), \qquad \chi = px,\tag{C.5}$$

and we require the UV boundary values

$$u \to 0: \qquad T(u) = U(u) = W_1(u) = W_2(u) = 1,\tag{C.6}$$

$$a(u) = \mu = \tilde{\mu}, \qquad \psi(u) = \lambda.\tag{C.7}$$

---

[11]Note that this definition is equivalent to $X = -i\mathfrak{g}\Psi$.

[12]When $k_y = 0$ we find $\mathfrak{g}_2 = 0$, so this decomposition is compatible with (A.9).

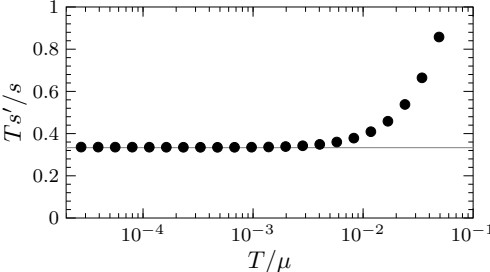

Figure 6: Scaling of the entropy at small temperature for the numerical Q-lattice background, obtained using pseudospectral relaxation. The gridline shows the zero-temperature scaling prediction of [18] (we have $\gamma = 1, \alpha = 1, c = 3$, which leads to $s \sim T^{1/3}$). It is clearly visible that below $T \sim 10^{-3}\mu$ our background series demonstrates the desired scaling and therefore approaches the non-AdS$_2$ fixed point discussed in section 3.

Note that by setting the UV values of all four metric components we define the dynamical gauge and moreover, we can use the remaining gauge freedom in order to relate $\tilde{\mu}$ to the chemical potential. Note also that the value $W_1 = 1$ ensures that the momentum of the lattice $p$ has the right relation to the corresponding boundary theory parameter. Finally, given the particular parameters we use in the action (16), and therefore the asymptotic behaviour of the scalar field $\phi$ (B.3), the value of the rescaled field $\psi$ is simply the source of the Q-lattice.

After all these preparations we have six elliptic equations for six functions with well-defined UV boundary conditions and the IR requirement that at the horizon the solutions are regular. The latter is achieved by simply expanding the equations near the horizon and solving for the derivatives of the fields in terms of their finite values, which gives a set of generalized Robin boundary conditions. We should also mention here that due to the increasingly steep behaviour of the fields at small temperatures (expected, of course, given that their profiles diverge at zero $T$, see appendix B) we find it very efficient to rescale the radial coordinate before applying the numerical solver:

$$u = 1 - (1 - z)^2. \tag{C.8}$$

This rescaling flattens out all the profiles at the horizon and improves the quality of the solutions.

For most of our calculations we use the Chebyshev grid for $z \in [0, 1]$ with $N = 80$. This allows us to reliably generate the Q-lattice backgrounds with temperatures as low at $T/\mu = 10^{-5}$. At these temperatures the solutions are already in the scaling region, characterized by the exponents evaluated in [18]. In particular, constructing the series of solutions at low temperature, we observe that for our choice of the parameters in (16) the entropy scales as $s = T^{1/3}$, see figure 6, which agrees perfectly with the prediction of [18] and our scaling treatment in (11), and ensures that we are studying the non-AdS$_2$ backgrounds in the correct scaling regime.

When solving for the fermionic spectral functions we write down the Dirac equation (2) in the metric ansatz (C.1), using the square roots of the metric functions as the values for the corresponding tetrad. For arbitrary momentum direction we get a set of four first order linear differential equations, which can be solved on the lattice in one step by `LinearSolve[]` in Wolfram Mathematica 13 [63]. The equations can be solved at $\omega = 0$, which we use through the text in order to study the spectral function at the Fermi surface.

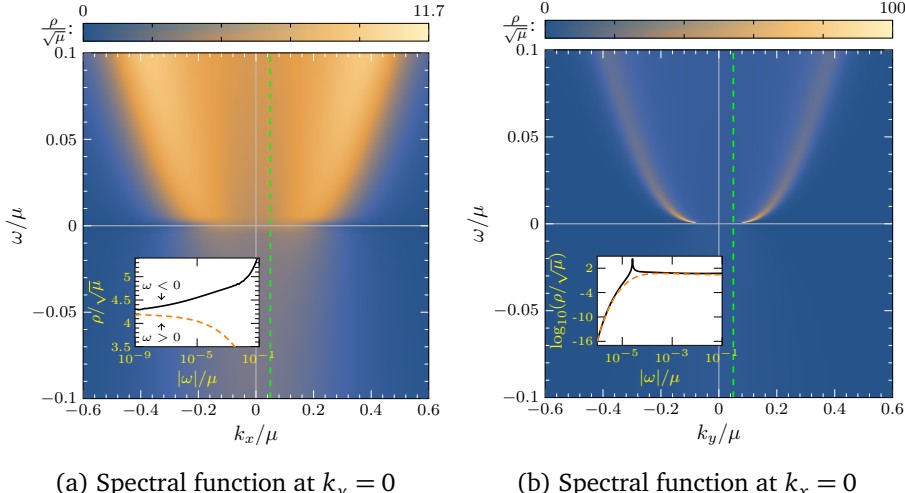

(a) Spectral function at $k_y = 0$      (b) Spectral function at $k_x = 0$

Figure 7: Slices of the fermionic spectral function $\rho(\omega, \vec{k}) = \mathrm{Tr}\,\mathrm{Im}\,G(\omega, \vec{k})$ in the anisotropic Q-lattice model (16) at $\lambda = \mu$, $p = 0.1\mu$ and zero temperature, for a bulk fermion with $m = 1/4$ and $q = 1$. The spectral function is finite at $\omega = 0$ when $\vec{k} = (k_x, 0)$, and vanishes at zero frequency when $\vec{k} = (0, k_y)$. This can clearly be seen in the insets, which show the spectral function along the example constant momentum cuts indicated by the vertical dashed green lines, i.e. at $k_x = 0.05\mu$ and at $k_y = 0.05\mu$ in the insets to figures 7a and 7a, respectively. The solid black curves in the insets show the spectral function for $\omega > 0$, while the dashed orange curves show the spectral function for $\omega < 0$.

# D    Additional numerical results

In this appendix we present a few extra numerical results for different values of the parameters in the anisotropic Q-lattice model considered in section 3. While in the main text we mainly had an explicit symmetry breaking source $\lambda = \mu$ and massless bulk fermion $m = 0$, here we present also results for a non-zero mass $m = 1/4$ in figures 7 and 8. We also show finite temperature results for weaker source $\lambda = 0.1\mu$, with both $m = 0$ and $m = 1/4$ in figure 9.

In all cases we observe qualitatively similar behaviour to that observed in section 3. The spectral function vanishes at $\omega = 0$ in the $y$ direction, which has positive scaling exponent $z_y = 3$, and stays finite at $\omega = 0$ in the $x$ direction, which has negative scaling exponent $z_x = -3$. These extra results support again our statement that the mechanism we discuss is to a large extent probe independent.

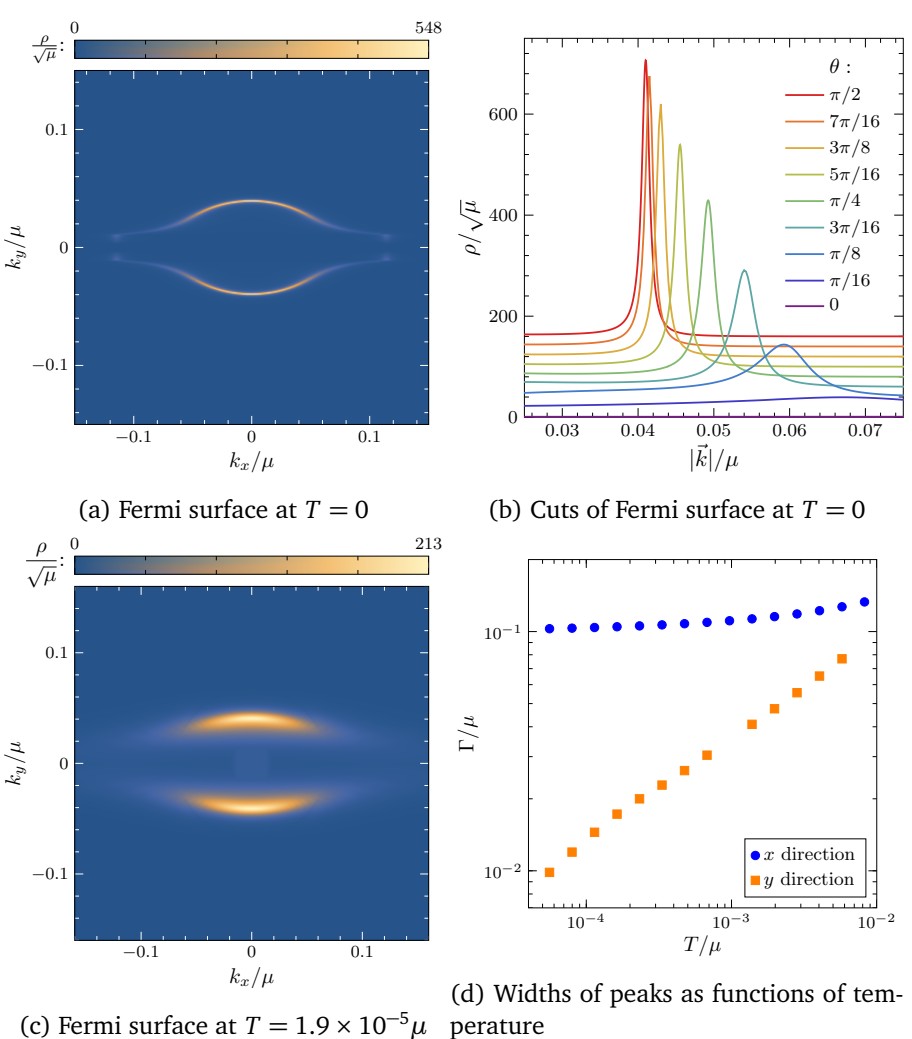

(a) Fermi surface at $T = 0$

(b) Cuts of Fermi surface at $T = 0$

(c) Fermi surface at $T = 1.9 \times 10^{-5}\mu$

(d) Widths of peaks as functions of temperature

Figure 8: Fermi surface plots for $\lambda = \mu$ and $p = 0.1\mu$, for a Dirac fermion with mass $m = 1/4$ and charge $q = 1$. **(a):** Spectral function at $T = 0$, at a small imaginary frequency $\mathrm{Im}\,\omega = 10^{-6}i\mu$ showing the shape of the Fermi surface. **(b):** Cuts of the Fermi surface at different angles. **(c):** Spectral function at zero frequency, at small non-zero temperature $T = 1.9 \times 10^{-5}\mu$. **(d):** The widths of the peaks in the spectral function in the $x$ and $y$ directions as functions of temperature.

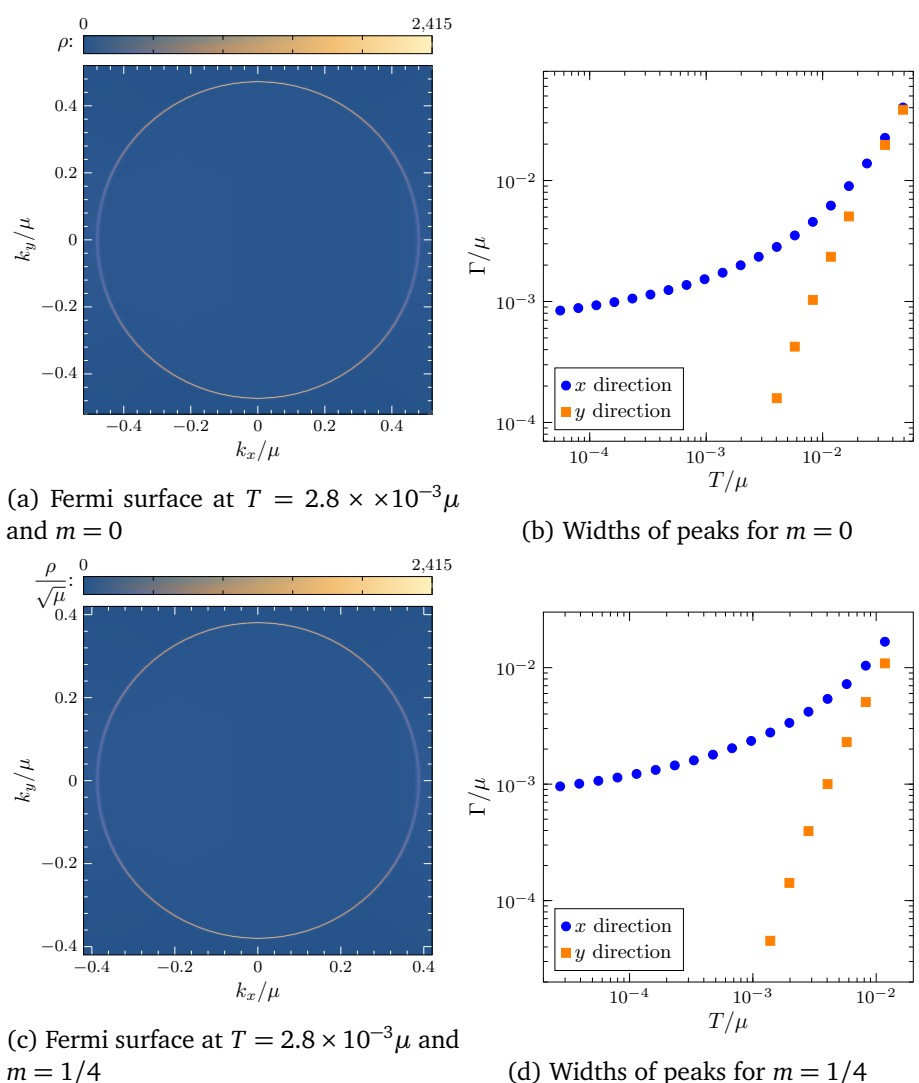

(a) Fermi surface at $T = 2.8 \times \times 10^{-3}\mu$ and $m = 0$

(b) Widths of peaks for $m = 0$

(c) Fermi surface at $T = 2.8 \times 10^{-3}\mu$ and $m = 1/4$

(d) Widths of peaks for $m = 1/4$

Figure 9: Fermi surface plots for $\lambda = 0.1\mu$ and $p = 0.1\mu$, for a Dirac fermion with charge $q = 1$. **(a):** Spectral function for $m = 0$ at zero frequency and a small temperature $T = 2.8 \times 10^{-3}\mu$ showing the shape of the Fermi surface. **(b):** The widths of the peaks in the spectral function in the $x$ and $y$ directions as functions of temperature for $m = 0$. **(c):** Spectral function for $m = 1/4$ at zero frequency and a small temperature $T = 2.8 \times 10^{-3}\mu$ showing the shape of the Fermi surface. **(d):** The widths of the peaks in the spectral function in the $x$ and $y$ directions as functions of temperature for $m = 1/4$.

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
