# Peer review of "Nodal-antinodal dichotomy from anisotropic quantum critical continua in holographic models"

_SciPost Physics, doi:SciPost Phys. 14, 161 (2023)_

## Round 2 · Referee Report · Anonymous (Referee 1) · 2023-2-17

Report

The paper argues that the Nodal-antinodal dichotomy could be understood in models of strongly correlated electrons via the holographic duality. The main idea is that the quantum critical continuum with spatial anisotropy can get imprinted on the fermion spectral function, leading to washing out the quasiparticle peak in one direction while leaving it intact in the perpendicular one. Using a holographic Q-lattice model with suitable parameters, the authors demonstrate how this effect emerges due to the quantitatively different scaling of the self-energy in different directions.

Although it is not clear if this idea can be used to understand experimental data, this manuscript contains non-trivial results potentially of interest to the applied holography community and condensed matter community. The paper is clearly written and well-organized. Therefore, I recommend the manuscript for publication without change in SciPost Physics.

---

## Round 2 · Referee Report · Hyun-Sik Jeong (Referee 2) · 2023-2-20

Report

This manuscript investigated the holographic fermion spectral function in presence of anisotropy in spatial dimensions. In particular, from the gravity perspective, the authors focus on how to engineer and understand the nodal-antinodal dichotomy on the Fermi surface in which the spectral function loses its quasi-particle peak in one of the spatial directions.

Using the exemplified gravity model, holographic Q-lattice model, the authors analyze the scaling behavior of IR geometry and find the analytic necessary conditions (or parameter range) relevant for the nodal-antinodal dichotomy in holography.

Furthermore, following the standard holographic method computing fermion spectral function in detail, the authors numerically verify that spectral function satisfying the necessary condition can mimic the nodal-antinodal behavior, which is also consistent with the given analytic argument from the IR green's function.

The holographic setup may be oversimplified in order to describe the real materials such as the one with the four-fold rotational symmetry or experimental results of the broken phase like underdoped cuprate superconductors. Nevertheless, the analysis in the manuscript is solid and pedagogical by the well organized explanations and calculations. Moreover, the results are novel, and allow for a further investigation of connection between the (anisotropy of) quantum critically and strongly correlated systems in a controllable setting. I believe the results are correct and of interest to the community, and so I can happily recommend publication in SciPost Physics with the suggestion of minor changes as follows.

1) On page 2: "..., as happens along the $k_a$ cut in the figure, ..." should be described with $k_n$ rather than $k_a$ in that the overlap happens in the $k_n$ direction in the figure 1.

2) On page 7: the argument of the functions ($\chi_{\pm}, \psi_{\pm}$) in Eq. (14) would be $\zeta$ rather than $r$.

3) On page 8: as demonstrated around Eq. (17), the chosen parameter $(z_x, z_y) = (-3, 3)$ corresponds to $\theta/\bar{z}=-1/3$ when both $\theta$ and $\bar{z}$ are divergent. Is it related with the semi-locally critical limit where the IR geometry is conformal to AdS ? If this is the case, I suspect that the given model, Eq. (16), may exhibit the linear resistivity in temperature, which is one of the major properties of the cuprate. Therfore, the semi-locally critical limit (if this is true in this model) may be another motivation to consider the model Eq.(16) in order to study the nodal-antinodal dichotomy in underdoped curate superconductors.

4) On page 24: $k_x$ in Eq. (67) may be replaced with $k_1$ in order to be consistent with the text below Eq. (69): "... the momentum of the lattice $k_1$ ...".

5) On page 27: when the fermion bulk mass $m$ is non-zero, the spectral function $\rho$ may not be a dimensionless quantity. Thus, the label for $\rho$ in Fig. 7 should be revised accordingly (for instance, $\frac{\rho}{\sqrt{\mu}}$). Also, in the same figure, the vertical dashed green lines seem to indicate $k_x=k_y=0.1 \mu$ contrary to the description in the caption.

---

## Round 2 · Referee Report · Anonymous (Referee 3) · 2023-2-23

Strengths

Clear and rigorous holographic calculation establishing an interesting and novel effect in the probe fermion spectrum. Good analytical control in a field where most of the work currently done is condemned to almost purely numerical analysis.

Weaknesses

Highly simplified model with no actual lattice. It is not clear how much of the results would remain valid in more realistic setups. One would like to understand better the meaning of the anisotropic positive-negative $z$ backgrounds and the relation to explicit holographic lattices.

Report

This is a very precise and well written paper about the effect of certain anisotropic backgrounds on the holographic Fermi surfaces. The basic idea is that isotropy breaking in the background geometry, in addition to specific scaling properties in the infrared (IR), leads to Fermi surface vanishing in certain directions. Simple but very clear and convincing analytical results are a particular strength of the paper.

As I mentioned, the paper is very precise and I have no major objections to the text as it is, just a few nitpicks. On the other hand, I feel that a few things should be added in order to have a more complete and convincing story. A list of questions (from minor to moare significant) follows.

(1) In eq. (14) the bispinor components $\chi_\pm,\psi_\pm$ are strictly speaking functions of $\zeta$, not $r$.

(2) What are the units for the spectral weight plots? In Figs. 2, 3 and 9(a,b) the intensity scale varies a lot even though for fixed $m$ we expect roughly the same overall scale no matter what the other quantities are. For the $m=1/4$ plots indeed it is natural that the scale changes.

(3) What would happen if we had a Q-lattice along both axes? I guess the same conclusions would hold unless the periods are the same?

(4) The claim in the Discussion (p. 13) that dipole coupling will not destroy the quantum critical continuum is somewhat misleading -- it is true that it will not change $\mathcal{G}$ at leading order but it will shift the pole in $G$ so that it can move the quasiparticle outside the continuum or destroy the Fermi surface everywhere.

(5) What exactly is $\Gamma$ in Fig. 3 and in the corresponding parts of the text? The explanation in the text is that it is the full width at half maximum but is that the right quantity to consider knowing that we have a pole superimposed on the continuum? Sure the way $\Gamma$ is defined it will behave as in Figs. 3(d), 8(d) and 9(b,d) but when computing the \emph{total} width we dump together the two contributions -- the pole is still there and if you would plot the spectrum in the complex $\omega$ plane you would see it along both directions.

This actually boils down to a more general question -- can you claim that Fermi surface vanishes if the pole is still there just drowned in the continuum?

(6) One puzzling aspect is the appearance of negative Lifshitz exponent $z_y$. Negative $z$ values can be found in some of the papers on effective holographic theories but certainly are not easy to understand. This should be shortly discussed in the paper -- I understand that a full analysis goes beyond the scope of the current work but a few sentences should be devoted to this so that the readers are not left puzzling on whether and how this makes sense.

(7) Smearing of the Fermi surface was seen (though not studied in detail) for example in [1910.01542] also in absence of the Q-lattice contribution (in that paper the authors, including one author of the current paper, interpolate between the Q-lattice and the true scalar lattice). However the anisotropy is there all the time as both contributions are only along the $x$-axis. So was that the same effect as here? Does the configuration in [1910.01542] satisfy the same requirements as found in section 2 here? Or maybe anisotropy by itself is enough? If you look for example at MDC plots in [1807.11730] and [2208.05920] you see weakening or vanishing of the Fermi surface along one or both axes, even though only the explicit (IR-irrelevant!) lattice is present there.

The only relatively significant weakness of this paper is that one can doubt the relevance of the mechanism demonstrated here for real-world systems, even for true holographic lattices. The authors offer convincing arguments that the mechanism considered in this paper is genuine and does not \emph{require} an explicit lattice. But the catch is: does it \emph{remain valid}? How do we know that the effect will not disappear upon hybridization?

In particular, the statement in the Discussion (p. 12) that the nodal-antinodal dichotomy "... doesn’t have to be explained in terms of fermiology, for example by resonance scattering between the different Fermi surfaces" is misleading -- it's true but the big question is -- does it remain valid also with the fermiology included?

For the reasons stated above, the relation to strange metal experiments is a bit oversold in the abstract, it is fair to say that the paper is inspired by this problem but not to state that "... the nodal-antinodal dichotomy in underdoped cuprate superconductors, can be reproduced ... via a holographic dual".

Overall, this is a valuable result which will certainly be an important step in further work. The paper should definitely be published with just a minor revision. The question is only, whether in SciPost Physics or in SciPost Physics Core, since the paper deals with a relatively simplified model and provides more of a technical result than a complete description of a physical effect. I'm perhaps a little bit more in favor of SciPost Physics as the paper is well written and likely to stimulate further work so it should be visible. Either way, it will be a useful resource for the community.

---

## Round 3 · Referee Report · Anonymous (Referee 3) · 2023-3-17

Report

The explanations and clarifications added in the text and discussed in the authors' response indeed make the formerly confusing points much clearer and more explicit. The discussion of the IR-irrelevant effects' dependence on the temperature and their disappearance as $T\to 0$ unlike the effect observed here is particularly important as a sharp criterion for distinguishing between the two. I thank the authors for their detailed and patient explanations (the text of the response will also be valuable for many readers, it's good that it will remain publicly visible).

Therefore, I very much recommend the publication of the paper in its current form.

---

## Round 3 · Author Response

Dear Editor We would like to thank the Referees for their careful analysis and the useful feedback. Here we provide our replies to the comments of the three referees. The summary of all changes (including minor ones) is attached.

Reply to Referee 1

We are happy to see that the Referee picks exactly the idea which we tried to express in our work and we thank them for the positive feedback.

Reply to Referee 2

We thank the Referee for the positive feedback, for pointing out the misprints and suggesting an interesting conceptual question.

(3)... both $\theta$ and $\bar{z}$ are divergent. Is it related with the semi-locally critical limit where the IR geometry is conformal to AdS ? If this is the case, I suspect that the given model, Eq. (16), may exhibit the linear resistivity in temperature, which is one of the major properties of the cuprate. ...

With $\bar{z}$ divergent but $z_{x,y}$ separately finite, the IR geometry is not conformal to AdS. We therefore do not expect the same semi-local critical behaviour as observed in isotropic models with divergent $z$. In particular, we anticipate that the IR scaling of the conductivity will be anisotropic and controlled by $z_x$ or $z_y$, depending on the direction of current flow.

We also corrected the misprints highlighted in the report points (1),(2),(4),(5)

Reply to Referee 3

We thank the Referee for the thorough report and for rising a few deep confusing points which we clarify in the revised version of the manuscript and which help to improve it. Here is our more detailed reply to the Referee's comments:

(2)What are the units for the spectral weight plots? ...

When $m=0$ the spectral function is dimensionless. In figure 2 the scale is fixed such that regions with $\rho > 100$ are cut off (this is done in order to make the rest of the structure of the spectral function visible). A comment on this has been added to the caption.

(3) What would happen if we had a Q-lattice along both axes? I guess the same conclusions would hold unless the periods are the same?

This is an interesting question which we plan to address in the follow ups. In the present work we consider the ''trivially'' anisotropic Q-lattice which is characterized by a single axion field. It is indeed interesting to study a ''non-trivially'' anisotropic model, where the two non-identical axion fields would be present and source the anisotropic deep IR geometry. This latter case is a bit more convoluted, but has been studied in detail in [1708.08822] and has been shown to lead to non-trivial anisotropy in the deep IR of the form (2.3) for some choices of the parameters. The advantage of this type of the Q-lattice is that the translation symmetry breaking is present in both directions and therefore one can study the non-trivial conductivity, besides the fermionic response.

We stress however, that the Q-lattice for us is just one of the ways to source deep IR anisotropy (this is discussed in the first paragraph of p.13), so while these more complex models are interesting to study, our main message is not affected: as long as there is any anisotropic scaling geometry of the type (2.3) in the deep IR, our mechanism for nodal-antinodal dichotomy works.

(4) The claim in the Discussion (p. 13) that dipole coupling will not destroy the quantum critical continuum is somewhat misleading -- it is true that it will not change G at leading order but it will shift the pole in G so that it can move the quasiparticle outside the continuum or destroy the Fermi surface everywhere.

It seems that there is a confusion here. By ''continuum'' we understand the finite density of the spectral function, which is not accounted for by a simple pole. In particular, the finite constant value of Im(G) in the example we consider is a continuum. When we discuss the existence of the continuum, we do it irrespective of whether the Fermi surface (the pole in the fermionic Green's function) exists or not. In case there is a pole, it will be broadened by the presence of continuum and this is an indirect way of observing the latter.

In this way our claim in the Discussion should not be misleading: the dipole coupling does not affect the features of the continuum. If the pole is not there or is moved away, this has no relevance to the features of continuum either. We rephrased this sentence now accordingly.

(5) What exactly is $\Gamma$ in Fig. 3 and in the corresponding parts of the text? The explanation in the text is that it is the full width at half maximum but is that the right quantity to consider knowing that we have a pole superimposed on the continuum? Sure the way $\Gamma$ is defined it will behave as in Figs. 3(d), 8(d) and 9(b,d) but when computing the total width we dump together the two contributions -- the pole is still there and if you would plot the spectrum in the complex $\omega$ plane you would see it along both directions.}

Indeed, $\Gamma$ here is the result of fitting the peaks in the numerical data, and we call it ``full width at half max'' (FWHM) in order to avoid extra theoretical bias.

It is related to the $Im \mathcal{G}$ as one can see from, i.e. eq.(7) in the following way. Consider this expression at zero $\omega$ as a function of $k$. In vicinity of the pole (at $k=k_f$) the denominator acquires a minimum value and can be parametrized as $c_{\pm} + d_{\pm} Re \mathcal{G}_{\pm} = v_f (k-k_f)$. The line shape in the momentum distribution curve (MDC) is then approximately Lorentzian with FWHM proportional to $Im \mathcal{G}_{\pm}(k_f, \omega=0)$.

One can also analyze the same line shape departing from the perturbative treatment of continuum as a contribution to self energy $\Sigma$ of the stable quasiparticle (with dispersion relation $\omega = v_f (k-k_f)$ and infinite lifetime). Assuming that self-energy is small one develops the perturabtive expansion and resums it using the Schwinger-Dyson equations. In this case the ''dressed'' fermionic 2-point function takes the form

$$ G = \frac{1}{\omega - vf (k-k_f) - \Sigma} $$
This form of the Green's function also leads to a Lorentzian shape of the peak in MDC width FWHM proportional to $Im \Sigma$. This is the reason why one usually refers to the width of the quasiparticle peak as the self energy. However, we should stress here that this perturbative interpretation loses sense in case when the ``self energy'' is not small, as it happens in our case when it is constant at the Fermi surface. Therefore we are forced to be careful with terminology and just call $\Gamma$ a FWHM. We added a comment about this in the end of the first paragraph of p.12.

This actually boils down to a more general question -- can you claim that Fermi surface vanishes if the pole is still there just drowned in the continuum?

This is indeed a subtle point and this is why we are trying to be careful with our statements throughout the paper. We usually discuss the ``existence of stable quasiparticle'' rather than the existence of the Fermi surface. The former can be clearly defined as the requirement that the self energy $\Sigma$ is much less then the frequency of the excitation (as we point out in the first par. of p.2 in the Introduction). If the stable quasiparticles exist, one can further define their Fermi surface. On the other hand, talking about Fermi surface in absence of stable quasiparticles doesn't seem to make much sense.

From the point of view of the analytic structure of Green's function in the complex $\omega$ plane, the strong contributions of $\Sigma$ would look like the extra poles or branch cuts close to the origin and therefore the single pole approximation (which allows one to prescribe a physical sense to the ``quasiparticle'' pole) fails.

(6) One puzzling aspect is the appearance of negative Lifshitz exponent $z_y$. Negative z values can be found in some of the papers on effective holographic theories but certainly are not easy to understand. This should be shortly discussed in the paper...

We added footnote to page 7 with some discussion of negative Lifshitz exponents, and a reference to [1401.5436].

(7) Smearing of the Fermi surface was seen (though not studied in detail) for example in [1910.01542] also in absence of the Q-lattice contribution (in that paper the authors, including one author of the current paper, interpolate between the Q-lattice and the true scalar lattice). However the anisotropy is there all the time as both contributions are only along the x-axis. So was that the same effect as here? Does the configuration in [1910.01542] satisfy the same requirements as found in section 2 here? Or maybe anisotropy by itself is enough? If you look for example at MDC plots in [1807.11730] and [2208.05920] you see weakening or vanishing of the Fermi surface along one or both axes, even though only the explicit (IR-irrelevant!) lattice is present there.

The crucial difference between the effect we describe here and the one in [1910.01542] is indeed the fact that the latter is due to irrelevant explicit lattice. I.e. it is visible at finite temperature, but as the temperature is lowered, the imaginary part of the IR Green's function reduces to the one of AdS-RN and the width of the peak would sharpen. In the context of irrelevant lattices this means that the difference between the peaks in the different directions is disappearing at low temperature. In our present case, as we highlight by the plots on Fig.3(d) etc, the width of the peak is finite even in zero temperature limit, and the difference between the widths of the peaks in the different directions grows as the temperature is lowered. This is totally due to IR-relevant deformation of the metric. In [1807.11730] and [2208.05920] the temperature dependence of the width is never studied and, similarly to the discussion above about the frequency dependence, it is not the value at some point, but the functional form of the self energy, which matters for the existence/non-existence of the stable quasiparticles. We added a reference to [2208.05920] and highlighted this difference more in the discussion in par.3 of p.11

(8) The only relatively significant weakness of this paper is that one can doubt the relevance of the mechanism demonstrated here for real-world systems, even for true holographic lattices. The authors offer convincing arguments that the mechanism considered in this paper is genuine and does not require an explicit lattice. But the catch is: does it remain valid? How do we know that the effect will not disappear upon hybridization?

This is indeed an interesting point which we address in the Discussion. As discussed above, our effect relies on the IR relevant deformation of the geometry. The most common in literature periodic lattices (due to the modulation of the chemical potantial) are IR irrelevant, so they would not lead to the same effect. However, in the already mentioned work [1910.01542] it has been shown that one can produce the periodic scalar lattice which will lead to quite a similar phenomenology as the homogeneous Q-lattice discussed here. We see no reason why the realistic periodic IR relevant scalar lattice would be impossible to produce by taking the scalar potentials as we used here and using the periodic scalars instead of Q-lattice along the lines of [1910.01542]. This is definitely a viable way forward and the important task for the future research, since so far we are not aware of the available constructions of IR relevant periodic lattices.

In this work we highlight the new mechanism of destroying quasiparticles and demonstrate it on an idealized example. To answer the Referee question of whether or not it will work for periodic IR relevant cases, one needs to construct those first.

In particular, the statement in the Discussion (p. 12) that the nodal-antinodal dichotomy "... doesn’t have to be explained in terms of fermiology, for example by resonance scattering between the different Fermi surfaces" is misleading -- it's true but the big question is -- does it remain valid also with the fermiology included?

This is easier to address. If by ``fermiology'' we understand the actual position and the shape of the Fermi surface, then, as mentioned above, our mechanism is not dependent on the position (and even existence) of the peak in the MDC. If the poles exist in the Green's function, then the peaks will get blurred by the quantum continuum in certain directions no matter what is the shape and the arrangements of the Fermi surfaces. In the other way, if the nature of the continuum in a system is somewhat different from the collective effect of the quasiparticles, then we would not expect the particular fermiology to affect the features of the continuum itself. This is certainly the case in holography, where the shapes of the Fermi surfaces are controlled by the features of the probe and do not affect the background (and therefore the IR geometry). In the real quantum systems this is getting more speculative, of course.

(9) For the reasons stated above, the relation to strange metal experiments is a bit oversold in the abstract, it is fair to say that the paper is inspired by this problem but not to state that "... the nodal-antinodal dichotomy in underdoped cuprate superconductors, can be reproduced ... via a holographic dual".

We agree with this point and rephrased the abstract with ''similar to the ....''.

With best Regards

The authors

---

## Round 3 · List of Changes

• The axis labels $k_n$ and $k_a$ were accidentally swapped in the figure, and in a couple of places in the text ($k_a$ should be the direction with the continuum). This has been corrected.
  • Equation (14) has been corrected.
  • $k_x$ in equation (67) and $k_1$ in the text below equation (69) have been changed to $p$, for consistency with the rest of the text. We have also removed the subscript from $\chi$ in equation (67).
  • The dimensions of $\rho$ in figure 7 have been corrected. The dashed green lines have been moved to their correct locations, $k = 0.05\mu$.
  • A comment on the scales of axes has been added to the caption of fig 2.
  • We add a comment on the Q-lattice in both directions in the footnote 5 on p.8
  • We rephrased the sentence in the end of the 4th paragraph of p.12 and removed ''destroyed''
  • We added a comment about the relation between $\Gamma$ and $Im \mathcal{G}$ in the end of the first paragraph of p.12
  • A footnote has been added to page 7 with some discussion of negative Lifshitz exponents, and a reference to [1401.5436].
  • We expanded the discussion in par.3 of p.11
  • We a rephrased the abstract with ''similar to the ....''.

---

## Editorial Decision

published